# WHEN WITNESSES DEFEND:
# A WITNESS GRAPH TOPOLOGICAL LAYER
# FOR ADVERSARIAL GRAPH LEARNING

## ABSTRACT

Capitalizing on the intuitive premise that shape characteristics are more robust to perturbations, we bridge adversarial graph learning with the emerging tools from computational topology, namely, persistent homology representations of graphs. We introduce the concept of witness complex to adversarial analysis on graphs, which allows us to focus only on the salient shape characteristics of graphs, yielded by the subset of the most essential nodes (i.e., landmarks), with minimal loss of topological information on the whole graph. The remaining nodes are then used as witnesses, governing which higher-order graph substructures are incorporated into the learning process. Armed with the witness mechanism, we design *Witness Graph Topological Layer (WGTL)*, which systematically integrates both local and global topological graph feature representations whose impact are in turn automatically controlled by the robust regularized topological loss. We derive the important stability guarantees of both local and global topology encodings and the associated robust topological loss, given the attacker's budget. We illustrate versatility of WGTL by its integration with GNNs and existing non-topological defense mechanisms. Our extensive experiments demonstrate that WGTL boosts the robustness of GNNs against a wide spectrum of adversarial attacks, leading to relative gains up to 18%.

## 1 INTRODUCTION

Graphs are ubiquitous data structures with applications in numerous knowledge domains: from structural representation of molecules in chemistry and material science to cryptocurrency transaction networks in finance. With their prevalence, it is important to learn effective graph representations and then apply them to solve downstream learning tasks. In present, the most widely adopted machinery for graph learning tasks is arguably Graph Neural Networks (GNNs) (Zhou et al., 2022). However, similar to the deep neural networks (DNN), GNNs exhibit vulnerability to adversarial attacks (Jin et al., 2021), i.e. small, often unnoticeable perturbations to the input graph might result in substantial degradation of GNN's performance in downstream tasks. In turn, compared to non-graph data, adversarial analysis of graphs is still in its infancy (Sun et al., 2022). Hence, developing robust GNN models that can resist a wide spectrum of adversarial attacks is of significant practical importance.

Presently, the three main strategies to defend GNNs against adversarial attacks are graph purification, adversarial training, and adversarial defense based neural architectures (Günnemann, 2022; Mujkanovic et al., 2022). These existing methods largely rely on the information at a node level while ignoring the higher-order, multi-scale properties of the graph structure, which are often the key behind the success of the learning task (Benson et al., 2018; Torres et al., 2021). These approaches also do not explicitly explore how to robustify GNNs by encoding adversarially resistant features.

In turn, in the last few years, we observe a spike of interest in the synergy of graph learning and Persistent Homology (PH) representations of graphs (Zhao & Wang, 2019; Carrière et al., 2020; Horn et al., 2022; Yan et al., 2022; Chen et al., 2022; Hajij et al.; Chen & Gel, 2023). PH representations enable us to glean intrinsic information about the inherent object shape. By shape here, we broadly understand properties which are invariant under continuous transformations such as twisting, bending, and stretching. This phenomenon can be explained by the important higher-order information, which

PH-based shape descriptors deliver about the underlying graph-structured data. This leads to an enhanced GNN performance in a variety of downstream tasks, such as link prediction, node and graph classification (Hofer et al., 2020; Carrière et al., 2020; Chen et al., 2021; Yan et al., 2021; Horn et al., 2022). Furthermore, in view of the invariance with respect to continuous transformations, intuitively we can expect that shape characteristics are to yield higher robustness to random perturbations and adversarial attacks. While this intuitive premise of robustness and its relationship with DNN architectures has been confirmed by some recent studies (Chen et al., 2021; Gebhart et al., 2019; Goibert et al., 2022), to the best of our knowledge, there yet exists no topological adversarial defense for GNNs.

In this work, we bridge this gap by merging adversarial graph learning with PH representations of graph-structured data. Our key idea is to leverage the concept of witness complex for graph learning. This allows us to enhance computational efficiency of the proposed topological defense, which is one of the primary bottlenecks on the way of wider adoption of topological methods, as well to reduce the impact of less important or noisy graph information. In particular, the goal of witness complex is to accurately estimate intrinsic shape properties of the graph using not all available graph information, but *only* a subset of the most representative nodes, called *landmarks*. The remaining nodes are then used as *witnesses*, governing which higher-order graph substructures shall be incorporated into the process of extracting shape characteristics and the associated graph learning task. Intuitively, the idea can be compared with focusing only on the shape of the object skeleton, which is invariant under deformations. This mechanism naturally results in the two main benefits. First, it allows us to drastically reduce the computational costs. Second, to extract only the most essential shape characteristics (i.e., skeleton shape). Our topological defense takes a form of the *Witness Graph Topological Layer (WGTL)* with three novel components: *local and global witness complex-based topological encoding*, *topology prior aggregation*, and *robustness-inducing topological loss*.

The *local witness complex-based features* encapsulate graph topology within the local node neighborhoods, while the *global witness complex-based features* describes global graph topology. Using only local topology prior to the loss function might be vulnerable to local attacks, while only global topology prior might be more susceptible to global attacks. To defend against both types of attacks, both local and global topology prior needs to be combined, thus motivating the design of the topology prior aggregator. Inspired by the results (Hu et al., 2019; Carriere et al., 2021), we *use the robust topological loss as a regularizer to a supervised loss for adversarially robust node representation learning*. This allows to control which shape features are to be included into the defense mechanism. Furthermore, given an attacker's budget, we theoretically derive the stability guarantees of both local and global topology encodings, and the associated topological loss. Figure 2 shows the schematic of the proposed components. The proposed WGTL is versatile as the proposed shape features can be readily integrated with any GNN architecture. Our extensive numerical experiments in conjunction with node classification tasks demonstrate that WGTL enhances performance of GNNs on clean graphs, as well as substantially improves their robustness again a broad range of adversarial attacks. Furthermore, we also demonstrate that WGTL can be incorporated to boost the robustness capabilities of existing graph defense mechanisms such as Pro-GNN (Jin et al., 2020).

Significance of our contributions can be summarized as follows:

- We propose the first approach that systematically bridges adversarial graph learning with persistent homology representations of graphs.
- We introduce a novel topological adversarial defense for graph learning, i.e. the *Witness Graph Topological Layer (WGTL)*, based on the notion of the witness complex. WGTL systematically integrates both local and global higher-order graph characteristics. Witness complex enables us to focus only on the most essential shape characteristics delivered by the landmark nodes, thereby reducing the computational costs and minimizing the impact of noisy graph information.
- We derive the stability guarantees of both local and global topology encodings and the robust topological loss, given an attacker's budget. These guarantees show that local and global encodings are stable to external perturbations, while the stability depends on the goodness of the witness complex construction.
- Our extensive experiments indicate that WGTL boosts robustness capabilities of GNNs across a wide range of local and global adversarial attacks, resulting in relative gains up to 18%. Furthermore, WGTL is smoothly integrable with other existing defenses, such as Pro-GNN, improving the relative performance up to 4.95%.

## 2 RELATED WORKS

**Adversarial Defenses for GNNs.** There are broadly three types of defenses: graph purification-based, adversarially robust training and adversarially robust architecture (Günnemann, 2022). Notable defenses that purify the input graph include Pro-GNN (Jin et al., 2020)(supervised) and SVD-GCN (Entezari et al., 2020)(unsupervised). The adversarially robust training methods, train by augmenting node features with gradients Kong et al. (2020), or datasets by generating worst-case perturbations (Xu et al., 2019). The goal is to train with the worst-case adversarial perturbations such that the learnt model weights become more robust against worst-case perturbation (Günnemann, 2022). However, adversarial training can not defend against more severe perturbation than the ones they were trained with. Better architectures such as VAE (Zhang & Ma, 2020), Bayesian uncertainty quantification (Feng et al., 2021), and Attention (Tang et al., 2020) have also been proposed for adversarial defense. However, none of these tools have explored the use of robust, graph topological features as prior knowledge for improved defense. Gabrielsson et al. (2020) designed a topology-driven attack on images and topological loss. However, this approach does not consider graph data and no adversarial defense is proposed. Among topology-driven defenses, GNNGuard (Zhang & Zitnik, 2020) considers graphlet degree vectors to encode node structural properties such as triangles and betweenness centrality. However, unlike the PH features used in WGTL, the graphlet approach is empirical, without theoretically guaranteed robustness properties.

**Persistent Homology with Witness Complexes.** While PH gets increasingly popular in ML applications, such as link prediction, classification, forecasting (Barannikov, 1994; Carlsson & Vejdemo-Johansson, 2021), the primary bottleneck to its wider adoption is its computational complexity. This has motivated the adoption of approximate simplicial representations such as the witness complexes (De Silva & Carlsson, 2004b). However, despite some initial promising results, including theoretical guarantees Arafat et al. (2019; 2020); Schönenberger et al. (2020); Poklukar et al. (2021); Chen & Gel (2023), application of witness complex in machine learning remains in its infancy. *By capitalizing on the computational efficiency and robustness capabilities of witness complex, this paper opens a path toward new witness complex-based adversarially robust learning on graphs.*

## 3 BACKGROUND: GRAPHS, PERSISTENT HOMOLOGY, WITNESS COMPLEXES

**Topology of Graphs.** $\mathcal{G} \triangleq (\mathcal{V}, \mathcal{E}, \boldsymbol{X})$ denotes an attributed graph. $\mathcal{V}$ is a set of $N$ nodes. $\mathcal{E}$ is a set of edges. $\boldsymbol{X} \in \mathbb{R}^{N \times F}$ is a node feature matrix, where each node corresponds to an $F$ dimensional feature. The adjacency matrix of $\mathcal{G}$ is a symmetric matrix $\boldsymbol{A} \in \mathbb{R}^{N \times N}$ such that $\boldsymbol{A}_{uv} \triangleq \omega_{uv}$, i.e., edge weight, if nodes $u$ and $v$ are connected and 0, otherwise. For unweighted graphs, we observe $\omega_{uv} = 1$. Furthermore, $\boldsymbol{D}$ represents the degree matrix of $\mathcal{G}$, such that $\boldsymbol{D}_{uu} \triangleq \sum_{v \in \mathcal{V}} \boldsymbol{A}_{uv}$ and 0, otherwise.

The central ideas leveraged in this paper are the local and global topology of a graph. The topology of a graph is defined by corresponding geodesic distance. The geodesic distance $d_{\mathcal{G}}(u, v)$ between a pair of vertices $u$ and $v \in \mathcal{V}$ is defined as the length of the shortest path between $u$ and $v$. The path length is defined as the sum of weights of the edges connecting the vertices $u$ and $v$. Endowed with the canonical metric induced by the geodesic distance $d_{\mathcal{G}} : \mathcal{V} \times \mathcal{V} \to \mathbb{R}^{\geq 0}$, a weighted simple graph $\mathcal{G}$ transforms into a metric space $(\mathcal{V}, d_{\mathcal{G}})$. For a given positive real number $\epsilon > 0$, the set of nodes that are no more than geodesic $\epsilon$ away from a given node determines the local topology of that node. When $\epsilon = \text{Diam}(\mathcal{G})$, i.e. the diameter of $\mathcal{G}$, we retrieve the global topology of the graph. Increasing $\epsilon$ from 1 to $\text{Diam}(\mathcal{G})$ allows us to retrieve the evolution of the inherent graph features, like connected components, cycles, voids, etc. (Edelsbrunner et al., 2002; Zomorodian, 2005).

**Persistent Homology.** In order to study the evolution of graph features, we take a Persistent Homology (PH)-based approach. Persistent homology is a method of computational topology that quantifies topological features by constructing simplicial complexes, i.e. a generalised graph with higher-order connectivity information such as cliques, over the dataset. For example, a unweighted subgraph of $\mathcal{G}$, say $\mathcal{G}_\alpha$, consisting of only edges with length more than $\alpha$ is a simplicial complex. The $d$-th homology group of a simplicial complex $\mathcal{G}_\alpha$ consists of its $d$-dimensional topological features, such as connected components ($d = 0$), cycles ($d = 1$), and voids ($d = 2$). Now, as we increase $\alpha$, we observe that more and more edges are removed from $\mathcal{G}$. Thus, we obtain a nested sequence of simplicial complexes $\mathcal{G}_{\alpha_1} \subseteq \ldots \subseteq \mathcal{G}_{\alpha_n} = \mathcal{G}$ for $\alpha_1 \leq \alpha_2 \leq \ldots \leq \alpha_n$. This nested sequence of simplicial complexes is called a *graph filtration* and $\alpha_i$'s denote the filtration values. To make the

process more systematic and informative, often an abstract simplicial complex $\mathscr{K}(\mathcal{G}_{\alpha_j})$ is constructed on each $\mathcal{G}_{\alpha_j}$, resulting in a *filtration* of complexes $\mathscr{K}(\mathcal{G}_{\alpha_1}) \subseteq \ldots \subseteq \mathscr{K}(\mathcal{G}_{\alpha_n})$. For a more detailed discussion on graph filtration, we refer to Hofer et al. (2020).

The key idea of PH is to choose multiple scale parameters $\alpha$ and study changes in topological features that occur to $\mathcal{G}$, which evolves with respect to $\alpha$. Equipped with the filtration of complexes, we can trace data shape patterns, i.e. the $d$ homology groups, such as independent components, holes, and cavities which appear and merge as scale $\alpha$ changes. For each topological feature $\rho$, we record the indices $b_\rho$ and $d_\rho$ of $\mathscr{K}(\mathcal{G}_{b_\rho})$ and $\mathscr{K}(\mathcal{G}_{d_\rho})$, where $\rho$ is first and last observed, respectively. We say that a pair $(b_\rho, d_\rho)$ represents the birth and death times of $\rho$, and $(d_\rho - b_\rho)$ is its corresponding lifespan (or *persistence*). In general, topological features with longer persistence are considered valuable, while features with shorter persistence are often associated with topological noise. The extracted topological information over the filtration $\{\mathscr{K}_{\alpha_j}\}$ is then represented in $\mathbb{R}^2$ as a *Persistence Diagram (PD)*, such that PD $= \{(b_\rho, d_\rho) \in \mathbb{R}^2 : d_\rho > b_\rho\} \cup \Delta$. $\Delta = \{(t,t)|t \in \mathbb{R}\}$ is the diagonal set containing points counted with infinite multiplicity. Another useful representation of persistent topological features is *Persistence Image* (PI) that vectorizes the persistence diagram with a Gaussian kernel and a piece-wise linear weighting function (Adams et al., 2017). Persistence images are deployed to make a classifier "topology-aware" and are known to be helpful in graph classification (Zhao & Wang, 2019; Rieck et al., 2020). Our methodology and experimental results shows topology-awareness can improve both the robustness and accuracy of graph classification.

**Witness Complexes.** There are multiple ways to construct an abstract simplicial complex $\mathscr{K}$ (Zomorodian, 2005). Due to its computational benefits, one of the widely adopted approaches is a *Vietoris-Rips complex* (VR). However, the VR complex uses the entire observed data to describe the underlying topological space, and thus, does not efficiently scale to large and noisy datasets (Zomorodian, 2010). In contrast, a *witness complex* captures the data shapes using only on a significantly smaller subset $\mathfrak{L} \subseteq \mathcal{V}$, called a set of *landmarks* (De Silva & Carlsson, 2004a). In turn, all other points in $\mathcal{V}$ are used as "witnesses" that govern the appearances of simplices in the witness complex. Arafat et al. (2020) demonstrate algorithms to construct landmark sets, their computational efficiencies, and stability of the induced *witness complex*. We leverage witness complex to scale to large graph datasets.

**Definition 1** (Weak Witness Complex (De Silva & Carlsson, 2004a))**.** *We call $w \in \mathcal{V}$ to be a weak witness for a simplex $\sigma = [v_0 v_1 \ldots v_l]$, where $v_i \in \mathcal{V}$ for $i = 0, 1, \ldots, l$ and $l \in \mathbb{N}$, with respect to $\mathfrak{L}$ if and only if $d_\mathcal{G}(w, v) \le d_\mathcal{G}(w, u)$ for all $v \in \sigma$ and $u \in \mathfrak{L} \setminus \sigma$. The weak witness complex $\mathrm{Wit}(\mathfrak{L}, \mathcal{G})$ of the graph $\mathcal{G}$ with respect to the landmark set $\mathfrak{L}$ has a node set formed by the landmark points in $\mathfrak{L}$, and a subset $\sigma$ of $\mathfrak{L}$ is in $\mathrm{Wit}(\mathfrak{L}, \mathcal{G})$ if and only if there exists a corresponding weak witness in the graph $\mathcal{G}$.*

## 4 LEARNING A ROBUST TOPOLOGY-AWARE GRAPH REPRESENTATION

The general idea is that encoding robust graph structural features as prior knowledge to a graph representation learning framework should induce a degree of robustness against adversarial attacks. Graph measures that capture global properties of the graph and measures that rely on aggregated statistics are known to be robust against small perturbations. Examples include degree distribution, clustering coefficients, average path length, diameter, largest eigenvalue and the corresponding eigenvector, certain centrality measures, e.g., betweenness and closeness centralities. However, these measures are not multiscale in nature. Therefore, they fail to encapsulate global graph structure at multiple levels of granularity. Many of them, e.g., degree distribution, clustering coefficients, only encode 1-hop or 2-hop information. Such information can be learned by a shallow GNN through message passing, rendering such features less useful as a prior. Features such as average path length and diameter are too coarse-scale (scalar-valued) and do not help a GNN to discern the nodes. Since existing robust graph features can not encode both local and global topological information at multiple scales, we introduce local and global topology encodings based on persistent homology as representations to the GNNs (Section 4.1). We also propose to use a topological loss as regularizer to learn topological features better (Section 4.2).

### 4.1 WITNESS GRAPH TOPOLOGICAL LAYER

Now, we describe the architecture of the Witness Graph Topological Layer (WGTL) (see Figure 1).

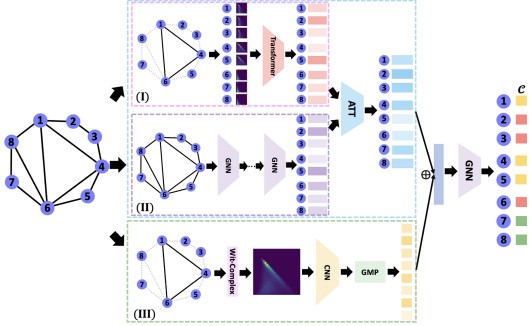

Figure 1: The overall architecture of Witness Graph Topological Layer.

**Component I: Local Topology Encoding.** *Local topology encoding* component of WGTL (see Figure 1) computes local topological features of every node in three steps. First, we choose a landmark set $\mathfrak{L}$ from the input graph $\mathcal{G}$. An important hyperparameter of the local topology encoding is the choice of the number of landmarks. Choosing too few landmarks would reduce the informativeness of the latent embedding. Choosing too many landmarks (i.e., $|\mathcal{V}|$), on top of being computationally expensive, might be redundant because the topological features of a neighboring node are likely to be the same. Secondly, we use the landmarks to construct an $\epsilon$-net of $\mathcal{G}$ (Arafat et al., 2020), i.e. a set of subgraphs $\{\mathcal{G}_l^\epsilon\}_{l \in \mathfrak{L}}$. Here, $\epsilon \triangleq \max_{l_1, l_2 \in \mathfrak{L}} 0.5 d_{\mathcal{G}}(l_1, l_2)$. We compute witness complex for each of these $\mathcal{G}_l^\epsilon$'s, and the corresponding persistence images $\mathrm{PI}(\mathrm{Wit}(\mathcal{G}_l^\epsilon))$. Finally, we attribute the PIs of the landmarks to each node in its $\epsilon$-cover and pass them through a vision transformer model to compute the local topology encoding, i.e. $\boldsymbol{Z}_{T_L} = \mathrm{Transformer}(\mathrm{PI}(\mathrm{Wit}(\mathcal{G}^\epsilon))_1, \ldots, \mathrm{PI}(\mathrm{Wit}(\mathcal{G}^\epsilon))_N)$. The local topology encoding $\boldsymbol{Z}_{T_L}$ is a latent embedding of local topological features of each node in $\mathcal{G}$.

When the attack model poisons the adjacency matrix, especially in the cases of global attacks, the local topological encodings are also implicitly perturbed. In Theorem 1, we show that local topological encodings are stable w.r.t. perturbations in the input graph. Specifically, if an attacker's budget is $\mathcal{O}(\delta)$, the encoded local topology is perturbed by $\mathcal{O}(C_\epsilon(\delta + \epsilon))$. The bound indicates the trade-off due to landmark selection. If we select less landmarks, computation becomes faster and we encode topological features of a larger neighborhood. But increase in $C_\epsilon$ yields less stable encoding. Whereas if we select more landmarks, we get more stable encoding but we loose informativeness of the local region and computational efficiency.

**Theorem 1** (Stability of the encoded local topology). *Let us denote the persistence diagram obtained from local topology encoding of $\mathcal{G}$ as $\mathrm{T}(\mathcal{G})$ (Figure 2). For any $p < \infty$ and $C_\epsilon$ being the maximum cardinality of the $\epsilon$-neighborhood created by the landmarks, we obtain that for any graph perturbation $\|\mathcal{G} - \mathcal{G}'\|_1 = \mathcal{O}(\delta)$ the final persistence diagram representation changes by $W_p(\mathrm{T}(\mathcal{G}), \mathrm{T}(\mathcal{G}')) = \mathcal{O}(C_\epsilon \delta)$, if we have access to Čěch simplicial complexes, and $W_p(\mathrm{T}(\mathcal{G}), \mathrm{T}(\mathcal{G}')) = \mathcal{O}(C_\epsilon(\delta + \epsilon))$, if Witness complex is used to compute the Local Persistence Images.*

**Component II: Graph Representation Learning.** The component II of WGTL deploys in cascade $M$ GNN layers with ReLU activation function and weights $\{\boldsymbol{\Theta}^{(m)}\}_{m=1}^M$. The representation learned at the $m$-th layer is given by $\boldsymbol{Z}_G^{(m+1)} = \mathrm{ReLU}(\widetilde{\boldsymbol{D}}^{-\frac{1}{2}} \widetilde{\boldsymbol{A}} \widetilde{\boldsymbol{D}}^{\frac{1}{2}} \boldsymbol{Z}_G^{(m)} \boldsymbol{\Theta}^{(m)})$. Here, $Z_G^{(0)} = \mathcal{G}$, $\widetilde{\boldsymbol{A}} = \boldsymbol{A} + \boldsymbol{I}$, and $\widetilde{\boldsymbol{D}}$ is the corresponding degree matrix.

**Component III: Global Topology Encoding.** The *global topological encoding* represents the global witness complex-based topological features of a graph (Component III in Figure 1). First, we use the input adjacency matrix to compute the lengths of all-pair shortest paths (geodesics) among the nodes. The topological space represented by the geodesic distance matrix is used to compute the global witness complex-based persistence image $\mathrm{PI}(\mathrm{Wit}(\mathcal{G}))$ of the graph (Arafat et al., 2020). Finally, the persistence image representation is encoded by a Convolutional Neural Network (CNN)-based model to obtain the *global topological encoding* $\boldsymbol{Z}_{T_G} \triangleq \xi_{\max}(\mathrm{CNN}(\mathrm{PI}(\mathrm{Wit}(\mathcal{G}))))$. Here, $\xi_{\max}(\cdot)$ denotes global max-pooling operation. The global topology encoding encapsulates the global topological features, such as equivalent class of connected nodes, cycles and voids in the graph.

The stability of global persistence diagram representation is a well-known classical result in persistence homology (Cohen-Steiner et al., 2005; Chazal et al., 2008). However, given an attacker's budget

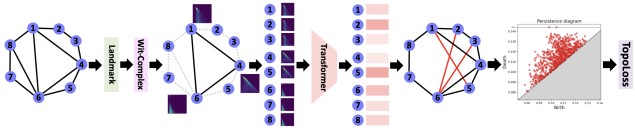

Figure 2: Illustration of Witness Complex-based topological regularizer $L_{Topo}$.

of $\delta$, the stability of the encoded global topology is an important result for the practical purposes of this paper. Theorem 1 shows that under a $\mathcal{O}(\delta)$ perturbation of the input graph, the global topology encoding is perturbed by $\mathcal{O}(\delta + \epsilon)$. Thus, the global topological encoding inherits the robustness property of persistent homology and thus, induces robust learning under adversarial attacks.

**Proposition 1** (Stability of the encoded global topology). *If the landmarks selected for the witness complex induce an $\epsilon$-net of the graph with $\epsilon > 0$, we obtain that for any graph perturbation $\|\mathcal{G} - \mathcal{G}'\|_1 = \mathcal{O}(\delta)$ the global persistence image representation changes by*

$$\|\mathrm{PI}(\mathrm{Wit}^{\mathrm{glob}}(\mathcal{G})) - \mathrm{PI}(\mathrm{Wit}^{\mathrm{glob}}(\mathcal{G}'))\|_\infty = \mathcal{O}(\delta + \epsilon),$$

*and it reduces to $\mathcal{O}(\delta)$, if we have access to the Cěch simplicial complexes for $\mathcal{G}$.*

**WGTL: Aggregating Global and Local Encodings.** We can aggregate the local and global topology encodings with the latent embedding of graph convolution layers in different ways. Figure 1 shows the approach that empirically provides the most effective defense against adversarial attacks (Ablation studies are deferred to Appendix C).

The aggregation of the three encodings is computed in two steps. First, to adaptively learn the intrinsic dependencies between learnt node embedding and latent local topological encodings, we utilize the attention mechanism to focus on the importance of task relevant components in the learnt representations, i.e. $(\alpha_G, \alpha_{T_L}) \triangleq \mathrm{Att}(\boldsymbol{Z}_H, \boldsymbol{Z}_{T_L})$. In practice, we compute attention coefficients as

$$\alpha_i = \mathrm{softmax}_i(\Upsilon_{\mathrm{Att}} \tanh(\boldsymbol{\Xi} \boldsymbol{Z}_i)) = \frac{\exp\left(\Upsilon_{\mathrm{Att}} \tanh\left(\boldsymbol{\Xi} \boldsymbol{Z}_i\right)\right)}{\sum_{j \in \{G, T_L\}} \exp\left(\Upsilon_{\mathrm{Att}} \tanh\left(\boldsymbol{\Xi} \boldsymbol{Z}_j\right)\right)},$$

where $\Upsilon_{\mathrm{Att}} \in \mathbb{R}^{1 \times d_{\mathrm{out}}}$ is a linear transformation, $\boldsymbol{\Xi}$ is the trainable weight matrix, and the softmax function is used to normalize the attention vector. Then, we obtain the final embedding by combining two embeddings $\boldsymbol{Z}_{\mathrm{AGG}} = \alpha_G \times \boldsymbol{Z}_G + \alpha_{T_L} \times \boldsymbol{Z}_{T_L}$. Finally, we combine the learnt embedding $\boldsymbol{Z}_{\mathrm{AGG}}$ with the latent global topological representation $\boldsymbol{Z}_{T_G}$, such that $\boldsymbol{Z}_{\mathrm{WGTL}} = \boldsymbol{Z}_{\mathrm{AGG}} \boldsymbol{Z}_{T_G}$. The node representation $\boldsymbol{Z}_{\mathrm{WGTL}}$ encapsulates both global and local topology priors. We call $\boldsymbol{Z}_{\mathrm{WGTL}}$ the *aggregated topological priors*. We feed $\boldsymbol{Z}_{\mathrm{WGTL}}$ into a graph convolutional layer and use a differentiable classifier (here we use a softmax layer) to make node classification. In the following, we show the stability of the aggregated topological priors.

**Proposition 2** (Stability of the aggregated topological encoding). *If the landmarks selected for the witness complex induce an $\epsilon$-net of the graph with $\epsilon > 0$ and $L_{\mathrm{GNN}}$ is the Lipschitz constant of the GNNs in Component II, then for a perturbation $\|\mathcal{G} - \mathcal{G}'\|_1 = \mathcal{O}(\delta)$, the encoding $\boldsymbol{Z}_{\mathrm{WGTL}}$ changes by*

$$\|\boldsymbol{Z}_{\mathrm{WGTL}}(\mathcal{G}) - \boldsymbol{Z}_{\mathrm{WGTL}}(\mathcal{G}')\|_1 = \mathcal{O}((C_\epsilon + L_{\mathrm{GNN}})(\delta + \epsilon)^2).$$

Proposition 2 shows that the final representations computed by WGTL is stable under adversarial attacks. The stability depends on the approximation trade-off induced by the landmark set and the Lipschitz stability of the GNN layers (Jia et al., 2023).

## 4.2 Topological Loss as a Regularizer

In Section 4.1, we propose using the aggregated topology encodings to predict node labels for downstream node classification tasks through a GNN backbone. In this case, we use a supervised loss $L_{supv}$ that facilitate learning the aggregated topology priors for classification. We empirically observe that our topology encoding already provides a certain degree of robustness (Appendix C).

However, the supervised loss function only explicitly enforces misclassification constraints on the defense model. It does not explicitly enforce any topological constraint such that the topological encodings themselves iteratively become more robust while training. Hence, for increased robustness,

we propose to use topological loss $L_{topo}$ that explicitly encodes the birth and death of the topological features in the auxiliary graph (ref. Figure 2) reconstructed from the transformer output. Specifically,

$$L_{topo,k}(\mathrm{T}(\mathcal{G})) \triangleq \sum_{i=1}^{m}(d_i - b_i)^p \left(\frac{d_i + b_i}{2}\right)^q, \tag{1}$$

where $m$ is the number of points in the persistence diagram of the auxiliary graph reconstructed from the transformer output and $k = \max\{p, q\}$. In practice, we use $k = 2$. Use of such topological loss was first proposed for image segmentation (Hu et al., 2019). Gabrielsson et al. (2020) uses it as a regularizer in designing GAN and adversarial attacks on images.

In contrast, we use it to induce stability in the encoding and to defend against adversarial attacks. The benefits of using the topological loss are two-fold: (i) **Persistent and stable feature selection:** Minimising $L_{topo,k}$ causes removal of topological features with smaller persistence, i.e., $(d_i - b_i)$. Thus, the regularizer acts as a sparsity-inducing feature selector. Thus, by minimising $L_{topo}$, we are training to learn latent representation such that only the most persistent features remain in the encoded local topology. Such features are known to be more stable and represent more robust structures of the graph, and (ii) **Robustness to local perturbations:** A localized attack perturbing certain nodes or edges is expected to appear as topological noise in the final persistent diagram, and thus, should exhibit a small persistence. Since minimizing $L_{topo}$ forces the local topology encodings to eliminate features with small persistence, using $L_{topo}$ as a regularizer with $L_{supv}$ induces robustness to local perturbations in final classification tasks.

Proposition 3 quantifies the stability of the topological regularizer $L_{topo,k}$ under any attack with perturbation budget $\mathcal{O}(\delta)$. Specifically, it shows that the stability depends on a trade-off between the maximum persistence of the final graph representation, $A_\Phi(\mathcal{G})$, in Figure 2, and the number of non-zero persistent features in the final encoding. Hence, it reflects our discussion above.

**Proposition 3** (Stability of $L_{topo}$). *Let us assume that the cardinality of any $\epsilon$-neighborhood of $\mathcal{G}$ grows polynomially, i.e. $C_\epsilon = \mathcal{O}(\epsilon^{-M})$ for an $M > 0$. If $m$ is the number of points in the persistence diagram, $2k = 2\max\{p, q\} > M$, and $A(\mathcal{G})$ is the auxiliary graph constructed from the local topology encodings (Fig. 2), $L_{topo,k}(\mathrm{T}(\mathcal{G}))$ is stable w.r.t. a perturbation of $\mathcal{G}$, i.e. $\|\mathcal{G} - \mathcal{G}'\|_1 = \delta$.*

$$\left|L_{topo,k}(\mathrm{T}(\mathcal{G})) - L_{topo,k}(\mathrm{T}(\mathcal{G}'))\right| = \mathcal{O}\left(k\left(\epsilon^{-4kM}\mathrm{Diam}(A(\mathcal{G})) + m\epsilon^{-2k}\mathrm{Diam}(\mathcal{G})^{2k}\right)\delta\right).$$

## 5 EXPERIMENTAL EVALUATION

We now evaluate the WGTL utility in the node classification tasks for clean and attacked graphs under a range of perturbation rates. Following Zügner & Günnemann (2019); Zügner et al. (2018); Jin et al. (2020), we validate the proposed approach on four benchmark datasets, namely Citeseer, Cora, Pubmed, and Polblogs. Our results show that using WGTL enhanced with the topological loss as regularizer improves node classification performance across all considered scenarios, thereby, validating the utility of robust topological features as priors. We also perform a one-sided two-sample $t$-test between the best result and the best performance achieved by the baseline, where * denote a significant, result. Note that, throughout our experiments, we use 0-dimensional topological features. We defer the dataset descriptions and implementation details to Appendix A.

**Adversarial Attacks: Local and Global.** We deploy four local and global poisoning attacks, with perturbation rate, i.e., the ratio of changed edges, from 0% to 25%, to evaluate the robustness of WGTL. We consider a fixed GCN without weight re-training as the surrogate for all attacks. As a local attack, we deploy nettack (Zügner et al., 2018). Nettack is a targeted attack that selects nodes without violating the degree distribution and feature co-occurrence of the original graph, and then, perturb the edges around them. *Due to the stability of WGTL and topological regularizer, we expect to be robust to such local attacks.* As global (non-targeted) poisoning attacks, we deploy mettack (Zügner et al., 2019), and two topological attacks, namely PGD (Xu et al., 2019) and Meta-PGD (Mujkanovic et al., 2022). Mettack treats the graph as a hyperparameter and greedily selects perturbations based on meta-gradient for node pairs until the budget is exhausted. PGD attack (Xu et al., 2019) adapts the well-known Projected Gradient Descent-based attack in adversarial ML to graphs. Recently, Meta-PGD is proposed (Mujkanovic et al., 2022) by applying PGD on the meta-gradients. It combines the effectiveness of mettack and PGD, and is shown to be the most effective topological attack at present. *Though global attacks are expected to be more challenging while using topological features, we demonstrate that WGTL still yields significant robustness.* Further details on attack implementations and attackers' budgets are in Appendix B.

Table 1: Performance (Accuracy±Std)under nettack.

| Dataset | Model | Number of perturbations per node | | | | | |
|---|---|---|---|---|---|---|---|
| | | 0 | 1 | 2 | 3 | 4 | 5 |
| Cora-ML | GCN | 82.87±0.93 | 82.53±1.06 | 82.08±0.81 | 81.69 ±0.59 | 81.26±0.88 | 80.69±0.81 |
| | GCN + WGTL (ours) | *83.83±0.55 | *83.41±0.87 | *82.74±0.65 | 82.06±0.82 | 81.64±0.55 | 80.98±0.67 |
| Citeseer | GCN | 71.56±0.63 | 71.37±0.46 | 71.14±0.47 | 70.49±0.59 | 70.05±0.35 | 69.77±0.50 |
| | GCN + WGTL (ours) | *72.56±0.82 | *72.27±0.72 | *72.10±0.78 | *71.81±0.88 | *70.95±0.64 | *70.95±1.05 |
| Pubmed | GCN | 81.70±0.30 | 81.64±0.33 | 81.50±0.29 | 81.48±0.29 | 81.29±0.43 | 81.03±0.27 |
| | GCN + WGTL (ours) | *83.93±0.06 | *83.91±0.12 | *83.84±0.11 | *83.70±0.11 | *83.68±0.09 | *83.55±0.13 |
| Polblogs | GCN | 94.40±1.48 | 88.91±1.06 | 85.39±0.86 | 83.03±0.87 | 81.20±1.63 | 79.39±0.96 |
| | GCN + WGTL (ours) | *95.95±0.15 | *91.47±0.33 | *89.10±0.69 | *88.98±0.83 | *88.63±1.20 | *87.14±0.70 |

Table 2: Performance (Accuracy±Std) under mettack.

| Dataset | Model | Perturbation Rate | | | | | |
|---|---|---|---|---|---|---|---|
| | | 0% | 5% | 10% | 15% | 20% | 25% |
| Cora-ML | GCN | 82.87±0.83 | 76.55±0.79 | 70.39±1.28 | 65.10±0.71 | 52.30±1.43 | 47.53±1.96 |
| | GCN + WGTL (ours) | *83.83±0.55 | *78.63±0.76 | *73.41±0.82 | *68.87±0.89 | *57.47±1.00 | *53.71±1.81 |
| Citeseer | GCN | 71.56±0.56 | 67.28±0.61 | 62.49±0.88 | 55.70±1.2 | 49.23±1.06 | 49.00±1.67 |
| | GCN + WGTL (ours) | *72.56±0.82 | *71.40±0.93 | *67.95±0.43 | *65.97±0.40 | *55.84±1.44 | *57.95±0.70 |
| Pubmed | GCN | 81.70±0.30 | 77.36±0.30 | 74.76±0.61 | 71.55±0.68 | 69.03±0.75 | 66.21±0.89 |
| | GCN + WGTL (ours) | *83.93±0.06 | *80.35±0.07 | *78.53±0.17 | *76.03±0.31 | *74.30±0.14 | *71.95±0.26 |
| Polblogs | GCN | 94.40±1.47 | 71.41±2.42 | 69.16±1.86 | 64.66±2.59 | 56.05±2.18 | 48.59±1.44 |
| | GCN + WGTL (ours) | *95.95±0.15 | *74.62±0.42 | *72.84±0.86 | *68.65±0.31 | *62.44±1.51 | *58.24±0.14 |

Table 3: Performance (Accuracy±Std) under PGD-attack.

| Dataset | Model | Perturbation Rate | | | | | |
|---|---|---|---|---|---|---|---|
| | | 0% | 5% | 10% | 15% | 20% | 25% |
| Cora-ML | GCN | 82.87±0.93 | 82.45±0.92 | 77.33±0.27 | 77.56 ±0.79 | 73.44±0.88 | 68.73±0.81 |
| | GCN + WGTL (ours) | *83.83±0.55 | *83.30±0.99 | *80.00±1.15 | 77.97±1.90 | *77.19±0.91 | *73.10±1.47 |
| Citeseer | GCN | 71.56±0.63 | 69.58±0.48 | 65.56±0.45 | 66.34±1.28 | 63.48±1.75 | 60.23±1.72 |
| | GCN + WGTL (ours) | *72.56±0.82 | *72.33±0.33 | *71.00±1.59 | *71.52±1.40 | *71.08±1.20 | *71.09±0.74 |
| Pubmed | GCN | 81.70±0.30 | 81.64±0.18 | 81.01±0.21 | 79.46±0.15 | 77.97±0.05 | 75.77±0.08 |
| | GCN + WGTL (ours) | *83.93±0.06 | *82.14±0.12 | *81.74±0.25 | *80.74±0.09 | *80.20±0.06 | *79.86±0.09 |
| Polblogs | GCN | 94.40±1.48 | 91.17±2.27 | 89.92±1.43 | 72.17±5.06 | 69.20±5.74 | 62.33±3.89 |
| | GCN + WGTL (ours) | *95.95±0.15 | 91.45±0.51 | 90.02±1.16 | *77.09±1.32 | *72.00±4.68 | *64.66±4.32 |

Table 4: Performance (Accuracy±Std) under Meta-PGD attack.

| Dataset | Model | Perturbation Rate | | | | | |
|---|---|---|---|---|---|---|---|
| | | 0% | 5% | 10% | 15% | 20% | 25% |
| Cora-ML | GCN | 82.87±0.93 | 79.30±0.86 | 76.26±0.92 | 74.09± 0.54 | 72.37±0.63 | 70.15±0.72 |
| | GCN + WGTL (ours) | *83.83±0.55 | 79.57±1.10 | 76.52±0.81 | 74.32±1.00 | 72.84±0.85 | *71.06±0.76 |
| Citeseer | GCN | 71.56±0.63 | 67.89±0.59 | 66.80±0.79 | 65.13±0.60 | 61.48±0.53 | 60.60±0.27 |
| | GCN + WGTL (ours) | *72.56±0.82 | *69.38±0.27 | *67.57±0.67 | 65.98±0.76 | *62.94±0.55 | *61.08±0.57 |
| Pubmed | GCN | 81.70±0.30 | 77.24±0.14 | 73.56±0.17 | 70.89±0.21 | 68.25±0.31 | 65.92±0.34 |
| | GCN + WGTL (ours) | *83.93±0.06 | *78.97±0.20 | *75.22±0.16 | *72.84±0.12 | *70.50±0.31 | *68.37±0.12 |
| Polblogs | GCN | 94.40±1.48 | 83.46±2.13 | 78.08±0.73 | 74.89±0.83 | 70.35±1.40 | 70.65±1.97 |
| | GCN + WGTL (ours) | *95.95±0.15 | *85.52±0.70 | *81.28±0.31 | *79.43±0.50 | *73.37±0.85 | *71.64±1.78 |

## 5.1 RESULTS: NODE CLASSIFICATION PERFORMANCE ON CLEAN AND ATTACKED GRAPHS

**Performance Evaluation of GCN + WGTL.** First, we evaluate the node classification accuracy of WGTL deployed with GCN as a classifier (Jia et al., 2023) against a local poisoning attack, i.e., nettack. Table 1 compares performance, i.e., mean and standard deviation of accuracies over 10 runs, of GCN with our GCN + WGTL, i.e., we use the GCN as a GNN backbone architecture. The best performance is highlighted in bold. Note that, despite being local, nettack preserves the degree distribution. Hence, the graph topology is expected to go through minimal change. Hence, intuitively nettack should diminish the informativeness of the local and global topology priors on the perturbed graphs. Despite that, we observe that GCN + WGTL still consistently outperforms the GCN across

different numbers of perturbations. Now, we compare performance of GCN against GCN+WGTL (i.e., we use the GCN as a GNN backbone architecture) against three global attacks, i.e. mettack, PGD-attack and Meta-PGD, and illustrate the results in Tables 2, 3 and 4, respectively. From these tables, we observe that: (i) GCN+WGTL consistently outperforms GCN, which proves the effectiveness of our proposed WGTL (for instance, our method achieves a relative gain of up to 13.00% on Cora-ML dataset under mettack), (ii) GCN+WGTL is often better or comparable to the backbone GCN in terms of standard deviation, indicating that the proposed components in WGTL do not affect the stability of the backbone model, (iii) The performance of our method, including that of the backbone GCN, deteriorates faster on Polblogs than on other datasets. This phenomenon can be explained by the fact that unlike other graph datasets, Polblogs does not have node features. That is, having informative node features can help GNN to differentiate between nodes and to learn meaningful representations despite changes in the graph structure. With node features lacking, the Polblogs has comparatively less resilience against graph structural perturbations injected by the various attacks. Such results were also observed by Jin et al. (2020), and (iv) WGTL is the most effective against mettack and the least effective against Meta-PGD. For instance, against mettack on Polblogs (25% perturbation rate), WGTL improves the accuracy by 20%, while against Meta-PGD, the accuracy gain is only 1.4%.

The main reasons are two-fold: (i) Meta-PGD is a stronger attack than the mettack (Mujkanovic et al., 2022), and (ii) global poisoning attacks target graph topology and are supposed to be more challenging for the proposed *topology-based* defense WGTL. Furthermore, we have performed ablation studies and refer to Appendix C for details.

Table 5: Node classification performance (Accuracy±Std) under mettack. Pro-GNN is used as a backbone GNN architecture.

| Dataset | Model | Perturbation Rate | | |
|---|---|---|---|---|
| | | 0% | 5% | 10% |
| Cora-ML | Pro-GNN | 82.98±0.23 | 80.14±1.34 | 71.59±1.33 |
| | Pro-GNN + WGTL (ours) | *$\mathbf{83.85 \pm 0.38}$ | *$\mathbf{81.90 \pm 0.73}$ | *$\mathbf{72.51 \pm 0.76}$ |
| Citeseer | Pro-GNN | 72.34±0.99 | 68.96±0.67 | 67.36±1.12 |
| | Pro-GNN + WGTL (ours) | *$\mathbf{72.83 \pm 0.94}$ | *$\mathbf{71.85 \pm 0.74}$ | *$\mathbf{70.70 \pm 0.57}$ |
| Pubmed | Pro-GNN | 87.33±0.18 | 87.25±0.09 | 87.20±0.12 |
| | Pro-GNN + WGTL (ours) | *$\mathbf{87.90 \pm 0.30}$ | *$\mathbf{87.77 \pm 0.08}$ | *$\mathbf{87.67 \pm 0.22}$ |

**Performance Evaluation of Pro-GNN+WGTL.** Table 5 illustrates the comparative performance of Pro-GNN (Jin et al., 2020) and Pro-GNN+WGTL on three citation networks under mettack. Similarly, Table 5 indicates that our Pro-GNN+WGTL is always better than the baseline on all datasets. For instance, we gain 0.68% - 4.96% of relative improvements on Cora-ML and Citeseer. The results reveal that WGTL enhances not only model expressiveness but also improves the robustness of the GNN-based model.

**Results: Computational Complexity.** Landmark selection (top-$|\mathfrak{L}|$ degree nodes) has complexity $\mathcal{O}(N \log(N))$. To compute witness topological features, one needs to compute (1) landmarks-to-witness distances costing $\mathcal{O}(|\mathfrak{L}|(N + |\mathcal{E}|))$ due to BFS-traversal from landmarks, (2) landmark-to-landmark distances costing $\mathcal{O}(|\mathfrak{L}|^2)$, and finally (3) persistent homology via boundary matrix construction and reduction (Edelsbrunner et al., 2002). Matrix reduction algorithm costs $\mathcal{O}(\zeta^3)$, where $\zeta$ is the number of simplices in a filtration. Overall, the computational complexity of computing witness topological feature on the graph is $\mathcal{O}(|\mathfrak{L}|(N + |\mathcal{E}|) + |\mathfrak{L}|^2 + \zeta^3)$. We observe that the average running times (i.e., training time per epoch) of WGTL (using GCN as the backbone architecture) on Cora-ML, Citeseer, Pubmed, and Polblogs are 3.86 s, 2.72 s, 6.97 s, and 0.78 s, respectively.

## 6 CONCLUSION

By harnessing the strengths of witness complex to efficiently learn topological representations based on the subset of the most essential nodes as skeleton, we have proposed the novel topological defense against adversarial attacks on graphs, WGTL. WGTL is versatile and can be readily integrated with any GNN architecture or another non-topological defense, leading to substantial gains in robustness. We have derived theoretical properties of WGTL, both at the local and global levels, and have illustrated its utility across a wide range of adversarial attacks. In the future, we plan to explore the utility of WGTL with respect to adversarial learning of time-evolving graphs and hypergraphs. Another interesting research direction is to investigate the linkage between the attacker's budget, number of landmarks, and topological attacks targeting the skeleton shape, that is, topological properties of the graph induced by the most important nodes (landmarks).

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

## A EXPERIMENTAL DETAILS

**Experimental Setup.** All experiments are run on a server with 32 Intel(R) Xeon(R) Silver 4110 CPU @ 2.10GHz processors, 256 GB RAM, and three NVIDIA V100 GPU cards with 32GB memory. All models are trained on a single GPU. The source code is avaiable at `https://www.dropbox.com/scl/fo/0oavxaw0vz2fjdtg1j76c/h?rlkey=b524wsqs60eci9zbryk91rj4a&dl=0`.

**Datasets.** Following Zügner & Günnemann (2019); Zügner et al. (2018); Jin et al. (2020), we validate the proposed approach on four benchmark datasets, including three citation graphs: Citeseer, Cora, and Pubmed, and one blog graph: Polblogs. For each graph, we randomly choose 10% of nodes for training, 10% of nodes for validation and the remaining 80% of nodes for testing. For each experiment, we report the average accuracy of 10 runs. Note that in the Polblogs graph node features are not available. Following Jin et al. (2020), we set the attribute matrix to $N \times N$ identity matrix.

Table 6: Dataset statistics: only the largest connected component (LCC) is considered.

| Dataset | #nodes (LCC) | #edges (LCC) | #classes | #features |
|---------|--------------|--------------|----------|-----------|
| Cora-ML | 2,485 | 5,069 | 7 | 1,433 |
| Citeseer | 2,110 | 3,668 | 6 | 3,703 |
| Pubmed | 19,717 | 44,338 | 3 | 500 |
| Polblogs | 1,222 | 16,714 | 2 | None |

**Landmark Selection for Local and Global Topology Encodings.** There are several approaches to selecting landmarks, e.g., random selection (De Silva & Carlsson, 2004a), max-min selection (De Silva & Carlsson, 2004a), $\epsilon$-net (Arafat et al., 2020) based and centrality-based selection (Chen & Gel, 2023). In our experiments, we select landmarks based on degree centrality. As shown by Chen & Gel (2023), doing so helps to improve the classification performance. On Cora-ML, Citeseer and Polblogs, we select 5% nodes, while on Pubmed, we select 2% nodes as landmarks. Each landmark creates its own cover consisting of a subset of nodes. A node $u$ belongs to the cover of a landmark $l$ if $l$ is nearest to $u$ among all the landmarks. Due to such landmark selection, the maximum values of $\epsilon$ are 2 for Citeseer and 3 for Cora-ML, Polblogs and Pubmed, and thus ensuring good stability of the encoded local and global topology.

## B    DETAILS OF ADVERSARIAL ATTACKS: CONFIGURATIONS AND BUDGETS

In this paper, we focus on four different local and global poisoning attacks to evaluate the robustness of our proposed WGTL, and consider a fixed GCN without weight re-training as the surrogate for all attacks. All attacks are considered under a non-adaptive setting, meaning it is assumed that the attacker can not adapt or interact with the model during the attack process. In all the poisoning attacks, we vary the perturbation rate, i.e., the ratio of changed edges, from 0% to 25% with a step of 5%.

**Global Poisoning Attacks.** Among global (non-targeted) poisoning attacks, we consider mettack (Zügner et al., 2019) and two different topological attacks: PGD (Xu et al., 2019) and its more recent adaptation Meta-PGD (Mujkanovic et al., 2022). Mettack treats the graph as a hyperparameter and greedily selects perturbations based on meta-gradient for node pairs until the budget is exhausted. PGD attack (Xu et al., 2019) uses the Projected Gradient Descent algorithm with the constraint $\|\boldsymbol{S}\|_0 \leq \delta$ to minimise attacker loss. Here, $\boldsymbol{S}$ is a binary symmetric matrix with $\boldsymbol{S}_{ij} = 1$ if the $(i, j)$-th entry of the Adjacency matrix is flipped by the attacker, and 0 otherwise. Recently, Mujkanovic et al. (2022) proposes to apply PGD on the meta-gradients to design attacks stronger than the greedy mettack. Meta-PGD has been shown to be more effective than mettack in many cases (Mujkanovic et al., 2022). Hence, we consider it a more challenging poisoning attack for the proposed method.

To perform mettack, we keep all the default parameter settings (e.g., $\lambda = 0$) following the original implementation (Zügner et al., 2019). For Cora-ML, Citeseer and Polblogs, we apply the Meta-Self variant of mettack since it is the most effective mettack variant, while for Pubmed, the approximate version of Meta-Self, A-Meta-Self, is applied to save memory and time (Jin et al., 2020). We perform the PGD attack with the CE-type attacker loss function. Following the implementation (Xu et al., 2019), we keep their default parameter settings, i.e., the number of iterations $T = 200$ and learning rate $\eta_t = 200/\sqrt{t}$. For Meta-PGD, we keep the same parameter settings as Mujkanovic et al. (2022), i.e., a learning rate of 0.01 and gradient clipping threshold of 1.

**Local Poisoning Attack.** Among local attacks, we use nettack (Zügner et al., 2018). Nettack is a targeted attack which first selects possible perturbation candidates not violating degree distribution and feature co-occurrence of the original graph. Then, it greedily selects, until the attacker's budget is exhausted, the perturbation with the largest score to modify the graph. The score function is the difference in the log probabilities of a target node.

Following Zügner et al. (2018), we vary the number of perturbations made on every targeted node from 1 to 5 with a step size of 1. Following Jin et al. (2020), the nodes in the test set with a degree $> 10$ are treated as target nodes. We only sample 10% of them to reduce the running time of nettack on Pubmed, while for other datasets, we use all the target nodes.

# C ABLATION STUDIES

To evaluate the contributions of the different components in our WTGL, we perform ablation studies on Cora-ML and Polblogs datasets under a global attack, i.e., mettack, and a local attack, i.e., nettack. We use GCN as the backbone architecture and consider three ablated variants: (i) GCN+Local Topology Encoding (LTE), (ii) GCN+Global Topology Encoding (GTE), and (iii) GCN+LTE+GTE+Topological Loss (TopoLoss) (i.e., GCN + WTGL).

The experimental results for mettack are shown in Tables 7 and 8. The experimental results for nettack are shown in Tables 9 and 10.

Consistent improvement from the backbone GCN while using GTE, LTE, and topological loss together suggest their importance in an individual as well as in an aggregated manner.

Table 7: Performance (Accuracy±Std) on Cora-ML under Mettack.

| Model | Cora-ML | | | | | |
|---|---|---|---|---|---|---|
| | 0% | 5% | 10% | 15% | 20% | 25% |
| GCN | 82.87±0.83 | 76.55±0.79 | 70.39±1.28 | 65.10±0.71 | 52.30±1.43 | 47.53±1.96 |
| GCN + LTE | 83.26±0.43 | 77.35±0.38 | 71.27±0.81 | 66.65±0.83 | 54.01±1.03 | 48.55±0.99 |
| GCN + GTE | 83.37±1.12 | 77.78±0.59 | 70.66±1.76 | 66.51±0.81 | 55.64±1.47 | 49.80±1.97 |
| GCN + LTE + GTE + TopoLoss | **83.83±0.55** | **78.63±0.76** | **73.41±0.82** | **68.87±0.89** | **57.47±1.00** | **53.71±1.81** |

Table 8: Performance (Accuracy±Std) under on Polblogs Mettack.

| Architecture | Polblogs | | | | | |
|---|---|---|---|---|---|---|
| | 0% | 5% | 10% | 15% | 20% | 25% |
| GCN | 94.40±1.47 | 71.41±2.42 | 69.16±1.86 | 64.66±2.59 | 56.05±2.18 | 48.59±1.44 |
| GCN + LTE | 95.34±0.73 | 72.27±1.07 | 72.02±0.97 | 68.14±0.99 | 59.37±2.72 | 56.16±1.71 |
| GCN + GTE | 95.07±0.09 | 72.78±0.57 | 73.14±1.59 | 68.22±0.11 | 61.37±1.13 | 58.24±1.13 |
| GCN + LTE + GTE + TopoLoss | **95.95±0.15** | **72.84±0.86** | **74.62±0.42** | **68.65±0.31** | **62.44±1.51** | **58.24±0.14** |

Table 9: Performance (Accuracy±Std) on Cora-ML under Nettack.

| Model | Cora-ML | | | | | |
|---|---|---|---|---|---|---|
| | 0 | 1 | 2 | 3 | 4 | 5 |
| GCN | 82.87±0.93 | 82.53±1.06 | 82.08±0.81 | 81.69 ±0.59 | 81.26±0.88 | 80.69±0.81 |
| GCN + LTE | 82.88±0.24 | 82.56 ±0.13 | 82.33±0.33 | 82.06±0.26 | 81.43±0.21 | 80.76±0.17 |
| GCN + GTE | 83.15±0.43 | 82.42±0.66 | 82.29 ±0.48 | 82.03±0.31 | 81.53±0.53 | 80.73±0.40 |
| GCN + LTE + GTE + TopoLoss | **83.83±0.55** | **83.41±0.87** | **82.74±0.65** | **82.06±0.82** | **81.64±0.55** | **80.98±0.67** |

Table 10: Performance (Accuracy±Std) on Polblogs under Nettack.

| Model | Cora-ML | | | | | |
|---|---|---|---|---|---|---|
| | 0 | 1 | 2 | 3 | 4 | 5 |
| GCN | 94.40±1.48 | 88.91±1.06 | 85.39±0.86 | 83.03±0.87 | 81.20±1.63 | 79.39±0.96 |
| GCN + LTE | 95.42±0.58 | 91.45±0.56 | 88.40±0.94 | 88.43±0.59 | 88.02±0.73 | 86.95±0.55 |
| GCN + GTE | 95.07±0.11 | 91.47 ±0.68 | 89.10±0.70 | 89.38 ±0.78 | 88.61±1.38 | 87.04±0.51 |
| GCN + LTE + GTE + TopoLoss | **95.95±0.15** | **91.47±0.33** | **89.10±0.69** | **88.98±0.83** | **88.63±1.20** | **87.14±0.70** |

## D  THEORETICAL ANALYSIS

### D.1  NOTATIONS

We dedicate this section to index all the notations used in this paper. Note that every notation is defined when it is introduced as well.

Table 11: Notations.

| | | |
|---|---|---|
| $\mathcal{G}$ | $\triangleq$ | A graph with a vertex set $\mathcal{V}$, an edge set $\mathcal{E}$, and features $\boldsymbol{X}$ |
| $N$ | $\triangleq$ | Cardinality of $\mathcal{V}$, i.e. the number of nodes |
| $F$ | $\triangleq$ | Dimension of features corresponding to each node |
| $\boldsymbol{A}$ | $\triangleq$ | Adjacency matrix of $\mathcal{G}$ |
| $\boldsymbol{D}$ | $\triangleq$ | Degree matrix of $\mathcal{G}$ |
| $d_{\mathcal{G}}(u, v)$ | $\triangleq$ | Geodesic distance between nodes $u$ and $v$ in graph $\mathcal{G}$ |
| $\mathrm{Diam}(\mathcal{G})$ | $\triangleq$ | The diameter of the graph $\mathcal{G}$ |
| $\mathfrak{L}$ | $\triangleq$ | The set of landmark nodes |
| $\epsilon$ | $\triangleq$ | Radius of the $\epsilon$-net induced on $\mathcal{G}$ by $\mathfrak{L}$ |
| $\mathcal{G}_l^\epsilon$ | $\triangleq$ | The $\epsilon$-neighborhood of the landmark $l$ in graph $\mathcal{G}$ |
| $C_\epsilon$ | $\triangleq$ | Maximum cardinality of the $\epsilon$-neighborhoods induced by the landmarks $\mathfrak{L}$ |
| $d_H(\mathcal{G}_1, \mathcal{G}_2)$ | $\triangleq$ | Hausdroff distance between graphs $\mathcal{G}_1$ and $\mathcal{G}_2$ |
| $W_p(\cdot, \cdot)$ | $\triangleq$ | $p$-Wasserstein distance |
| $\mathrm{PD}(\mathscr{K}(\mathcal{G}))$ | $\triangleq$ | Persistence diagram of the $\mathscr{K}$ simplicial complexes computed on a graph $\mathcal{G}$. |
| | | $\mathscr{K}$ can be Cĕch, Vietoris-Rips or Witness complex of dimension $d \in \mathbb{Z}_{\geq 0}$. |
| $\mathrm{PI}(\mathscr{K}(\mathcal{G}))$ | $\triangleq$ | Persistence image of the $\mathscr{K}$ simplicial complexes computed on a graph $\mathcal{G}$. |
| | | In our analysis, $\mathcal{K}$ can be Cĕch, Vietoris-Rips or Witness complex of $d \in \mathbb{Z}_{\geq 0}$. |
| $\boldsymbol{A}(\mathcal{G})$ | $\triangleq$ | The adjacency matrix constructed from the local topology encoding $\boldsymbol{Z}_{T_L}$ of the nodes |
| $\mathrm{T}(\mathcal{G})$ | $\triangleq$ | Persistence diagrams of dimension $d \in \mathbb{Z}_{\geq 0}$ constructed from $A(\mathcal{G})$ |
| $\boldsymbol{A}^{glob}(\mathcal{G})$ | $\triangleq$ | The adjacency matrix constructed from the global topology encoding $\boldsymbol{Z}_{T_G}$ of the nodes |

### D.2  PROPERTIES OF THE LOCAL WITNESS COMPLEXES

**Theorem 2** (Properties of Local Witness Complexes (Arafat et al., 2020))**.** .

1. **Finiteness of the landmark set.** The cardinality of the landmark set $\mathfrak{L}$ is $\left(\dfrac{\mathrm{Diam}(\mathcal{G})}{\epsilon}\right)^{\mathcal{O}\left(\log \frac{|\mathcal{V}|}{\epsilon}\right)}$. Here, $\epsilon \triangleq \max_{u,v \in \mathfrak{L}} \frac{1}{2} d(u, v)$, and is a tunable parameter.

2. **Stability of the landmark set.** The Hausdorff distance denoted as $d_H(\mathcal{V}, \mathfrak{L})$ between connected weighted graph $(\mathcal{V}, d_{\mathcal{G}})$ and its $\epsilon$-net induced subspace $(\mathfrak{L}, d_{\mathfrak{L}})$ is at most $\epsilon$, where $\mathfrak{L} \subseteq \mathcal{V}$ is the set of landmarks.

3. **3-approximation of Vietoris-Rips.** For any $\alpha > 2\epsilon$,

$$\mathrm{VR}_{\alpha/3}(\mathfrak{L}) \subseteq \mathrm{Wit}_\alpha(\mathcal{V}, \mathfrak{L}) \subseteq \mathrm{VR}_{3\alpha}(\mathfrak{L})$$
$$\implies W_\infty(\mathrm{PD}_{>2\epsilon}(\mathrm{VR}), \mathrm{PD}_{>2\epsilon}(\mathrm{Wit})) \leq 3\log 3$$
$$\implies W_\infty(\mathrm{PD}(\mathrm{VR}), \mathrm{PD}(\mathrm{Wit})) \leq 3\log 3 + 2\epsilon$$

### D.3 STABILITY OF THE TOPOLOGICAL REPRESENTATIONS

**Remark 1.** *Here, for building Persistence Images, we use Gaussian kernel with variance $\sigma$ and a weighting function $w$, such that $|\nabla w| = 1$ for graphs.*

**Theorem 1** (Stability of the Final PD in Figure 2)**.** *Let us assume $p < \infty$ and $C_\epsilon$ is the maximum size of the $\epsilon$-neighbourhood created by the landmarks. Let us denote the persistence diagram obtained from local topology encoding of $\mathcal{G}$ as $\mathrm{T}(\mathcal{G})$ (Figure 2).*

*(a) If Cěch complex is used to compute the local persistence images around each landmark, we obtain that for any graph perturbation $\|\mathcal{G} - \mathcal{G}'\|_1 = \mathcal{O}(\delta)$ the final persistence diagram representation changes by*

$$W_p(\mathrm{T}(\mathcal{G}), \mathrm{T}(\mathcal{G}')) = \mathcal{O}(C_\epsilon \delta). \tag{2}$$

*(b) If Witness complex is used to compute the local persistence images around each landmark, we obtain that for any graph perturbation $\|\mathcal{G} - \mathcal{G}'\|_1 = \mathcal{O}(\delta)$ the final persistence diagram representation changes by*

$$W_p(\mathrm{T}(\mathcal{G}), \mathrm{T}(\mathcal{G}')) = \mathcal{O}(C_\epsilon(\delta + \epsilon)). \tag{3}$$

*Proof.* In the following, we prove the two parts of this theorem.

*(a) Cěch Complex.*

$$
\begin{aligned}
W_p(\mathrm{T}(\mathcal{G}), \mathrm{T}(\mathcal{G}')) &\underset{(a)}{\leq} W_\infty(\mathrm{T}(\mathcal{G}), \mathrm{T}(\mathcal{G}')) \\
&\underset{(b)}{\leq} \|A(\mathcal{G}) - A(\mathcal{G}')\|_\infty \\
&\underset{(c)}{\leq} \|A(\mathcal{G}) - A(\mathcal{G}')\|_1 \\
&\underset{(d)}{\leq} \sum_{i=1}^{|\mathcal{V}|} \|\mathrm{PI}(\mathrm{Cech}(\mathcal{G}_i) - \mathrm{PI}(\mathrm{Cech}(\mathcal{G}'_i)_i\|_1 \\
&\underset{(e)}{=} \sum_{l=1}^{|\mathfrak{L}|} |\mathcal{G}_l^\epsilon| \times \|\mathrm{PI}(\mathrm{Cech}(\mathcal{G}_l^\epsilon) - \mathrm{PI}(\mathrm{Cech}(\mathcal{G}'_l^\epsilon)\|_1 \\
&\underset{(f)}{\leq} \left(\sqrt{5} + \sqrt{\frac{10}{\pi}} \frac{1}{\sigma}\right) \sum_{l=1}^{|\mathfrak{L}|} |\mathcal{G}_l^\epsilon| \times W_1(\mathrm{PD}(\mathrm{Cech}(\mathcal{G}_l^\epsilon)), \mathrm{PD}(\mathrm{Cech}(\mathcal{G}'_l^\epsilon))) \\
&\leq \left(\sqrt{5} + \sqrt{\frac{10}{\pi}} \frac{1}{\sigma}\right) \sum_{l=1}^{|\mathfrak{L}|} |\mathcal{G}_l^\epsilon| \times W_\infty(\mathrm{PD}(\mathrm{Cech}(\mathcal{G}_l^\epsilon)), \mathrm{PD}(\mathrm{Cech}(\mathcal{G}'_l^\epsilon))) \\
&\underset{(g)}{\leq} \left(\sqrt{5} + \sqrt{\frac{10}{\pi}} \frac{1}{\sigma}\right) \sum_{l=1}^{|\mathfrak{L}|} |\mathcal{G}_l^\epsilon| \times \|\mathcal{G}_l^\epsilon - \mathcal{G}'_l^\epsilon\|_\infty \\
&\leq \left(\sqrt{5} + \sqrt{\frac{10}{\pi}} \frac{1}{\sigma}\right) \sum_{l=1}^{|\mathfrak{L}|} |\mathcal{G}_l^\epsilon| \times \|\mathcal{G}_l^\epsilon - \mathcal{G}'_l^\epsilon\|_1 \\
&\underset{(h)}{\leq} \left(\sqrt{5} + \sqrt{\frac{10}{\pi}} \frac{1}{\sigma}\right) \max_l |\mathcal{G}_l^\epsilon| \, \|\mathcal{G} - \mathcal{G}'\|_1 \\
&= \mathcal{O}(C_\epsilon \|\mathcal{G} - \mathcal{G}'\|_1).
\end{aligned}
$$

Step (a) is true due to the fact that $W_p(x, y) \leq W_q(x, y)$ for $0 < p \leq q$ and for all $x, y$.

Step (b) is due to the stability theorem of persistence diagrams (Cohen-Steiner et al., 2005). Here, $A(\mathcal{G})$ represents the adjacency matrix constructed from the local topology encoding $\mathbf{Z}_{T_L}$ of the nodes, and $\mathrm{T}(\mathcal{G})$ is the persistence diagrams of a fixed dimension $d \in \mathbb{Z}_{\geq 0}$ constructed from $A(\mathcal{G})$.

Step (c) is true as $l_\infty$ norm is less than $l_1$ norm between two vectors.

Step (d) is true due to 1-Lipschitzness of the transformation from the local persistence images to the final adjacency matrices computed using Local Topology Encodings.

Equality (e) is due to the fact that the persistence images for all nodes $i \in \mathcal{G}_l^\epsilon$ are the same.

Inequality (f) is a direct consequence of the stability theorem of the persistence images (Adams et al., 2017, Theorem 10). Here, for building Persistence Images, we use Gaussian kernel with variance $\sigma$ and a weighting function $w$, such that $|\nabla w| = 1$ for graphs.

Inequality (g) is an application of the stability theorem of persistence diagrams (Cohen-Steiner et al., 2005) on each of the $\epsilon$-neighborhoods of the landmarks.

Inequality (h) is true due to the fact that $\sum_{i=1}^m a_i b_i \leq \left( \max_{i \in \{1,\dots,m\}} a_i \right) \sum_{i=1}^m b_i$ if $0 \leq a_i, b_i < \infty$.

*(b) Witness Complex.*

$$
\begin{aligned}
W_p(\mathrm{T}(\mathcal{G}), \mathrm{T}(\mathcal{G}')) &\leq W_\infty(\mathrm{T}(\mathcal{G}), \mathrm{T}(\mathcal{G}')) \\
&\leq \|A(\mathcal{G}) - A(\mathcal{G}')\|_\infty \\
&\leq \sum_{l=1}^{|\mathfrak{L}|} |\mathcal{G}_l^\epsilon| \times \|\mathrm{PI}(\mathrm{Wit}(\mathcal{G}_l^\epsilon)) - \mathrm{PI}(\mathrm{Wit}(\mathcal{G}'_l^\epsilon))\|_1 \\
&\leq \left( \sqrt{5} + \sqrt{\frac{10}{\pi}} \frac{1}{\sigma} \right) \sum_{l=1}^{|\mathfrak{L}|} |\mathcal{G}_l^\epsilon| \times W_1(\mathrm{PD}(\mathrm{Wit}(\mathcal{G}_l^\epsilon)), \mathrm{PD}(\mathrm{Wit}(\mathcal{G}'_l^\epsilon))) \\
&\leq \left( \sqrt{5} + \sqrt{\frac{10}{\pi}} \frac{1}{\sigma} \right) \sum_{l=1}^{|\mathfrak{L}|} |\mathcal{G}_l^\epsilon| \times W_\infty(\mathrm{PD}(\mathrm{Wit}(\mathcal{G}_l^\epsilon)), \mathrm{PD}(\mathrm{Wit}(\mathcal{G}'_l^\epsilon))) \\
&\underset{(i)}{\leq} \left( \sqrt{5} + \sqrt{\frac{10}{\pi}} \frac{1}{\sigma} \right) \sum_{l=1}^{|\mathfrak{L}|} |\mathcal{G}_l^\epsilon| \times \left( W_\infty(\mathrm{PD}(\mathrm{VR}(\mathcal{G}_l^\epsilon)), \mathrm{PD}(\mathrm{VR}(\mathcal{G}'_l^\epsilon))) + 6\log 3 + 4\epsilon \right) \\
&\underset{(j)}{\leq} \left( \sqrt{5} + \sqrt{\frac{10}{\pi}} \frac{1}{\sigma} \right) \sum_{l=1}^{|\mathfrak{L}|} |\mathcal{G}_l^\epsilon| \times \left( 2\|\mathcal{G}_l^\epsilon - \mathcal{G}'_l^\epsilon\|_\infty + 6\log 3 + 4\epsilon \right) \\
&\leq \left( \sqrt{5} + \sqrt{\frac{10}{\pi}} \frac{1}{\sigma} \right) \sum_{l=1}^{|\mathfrak{L}|} |\mathcal{G}_l^\epsilon| \times \left( 2\|\mathcal{G}_l^\epsilon - \mathcal{G}'_l^\epsilon\|_1 + 6\log 3 + 4\epsilon \right) \\
&\leq \left( \sqrt{5} + \sqrt{\frac{10}{\pi}} \frac{1}{\sigma} \right) \max_l |\mathcal{G}_l^\epsilon| \; (2\|\mathcal{G} - \mathcal{G}'\|_1 + 6\log 3 + 4\epsilon) \\
&= \mathcal{O}(C_\epsilon \; (\|\mathcal{G} - \mathcal{G}'\|_1 + \epsilon)).
\end{aligned}
$$

Most of the calculations above follow similar rationale as the analysis for the Cĕch complex except steps (i) and (j).

Step (i) is a consequence of Theorem 2.3.

Step (j) holds true due to the stability theorem of persistence diagrams constructed from Vietoris-Rips complex (Chazal et al., 2009). □

**Proposition 1** (Stability of the Global PD in Figure 1 with Witness Complex). *If the landmarks selected for the Witness complex induce an $\epsilon$-net of the graph with $\epsilon > 0$, we obtain that for any graph perturbation $\|\mathcal{G} - \mathcal{G}'\|_1 = \mathcal{O}(\delta)$ the global persistence image representation changes by*

$$\|\mathrm{PI}(\mathrm{Wit}^{\mathrm{glob}}(\mathcal{G})) - \mathrm{PI}(\mathrm{Wit}^{\mathrm{glob}}(\mathcal{G}'))\|_\infty = \mathcal{O}(\delta + \epsilon), \tag{4}$$

*and $\|\mathrm{PI}(\mathrm{Cech}^{\mathrm{glob}}(\mathcal{G})) - \mathrm{PI}(\mathrm{Cech}^{\mathrm{glob}}(\mathcal{G}'))\|_\infty = \mathcal{O}(\delta)$, if we have access to the Cĕch simplicial complexes for $\mathcal{G}$.*

*Proof.* $\mathrm{PI}(\mathrm{Wit}^{\mathrm{glob}}(\mathcal{G}))$ refers to the global persistence images computed from the witness complex of $\mathcal{G}$ (ref. Figure 1).

$$
\begin{aligned}
& \|\mathrm{PI}(\mathrm{Wit}^{\mathrm{glob}}(\mathcal{G})) - \mathrm{PI}(\mathrm{Wit}^{\mathrm{glob}}(\mathcal{G}'))\|_\infty \\
& \leq \|\mathrm{PI}(\mathrm{Wit}^{\mathrm{glob}}(\mathcal{G})) - \mathrm{PI}(\mathrm{Wit}^{\mathrm{glob}}(\mathcal{G}'))\|_1 \\
& \underset{(a)}{\leq} \left(\sqrt{5} + \sqrt{\frac{10}{\pi}}\sigma\right) W_1\left(\mathrm{PD}(\mathrm{Wit}^{\mathrm{glob}}(\mathcal{G})), \mathrm{PD}(\mathrm{Wit}^{\mathrm{glob}}(\mathcal{G}'))\right) \\
& \leq \left(\sqrt{5} + \sqrt{\frac{10}{\pi}}\frac{1}{\sigma}\right) W_\infty\left(\mathrm{PD}(\mathrm{Wit}^{\mathrm{glob}}(\mathcal{G})), \mathrm{PD}(\mathrm{Wit}^{\mathrm{glob}}(\mathcal{G}'))\right) \\
& \underset{(b)}{\leq} \left(\sqrt{5} + \sqrt{\frac{10}{\pi}}\frac{1}{\sigma}\right)\left(W_\infty\left(\mathrm{PD}(\mathrm{VR}^{\mathrm{glob}}(\mathcal{G})), \mathrm{PD}(\mathrm{VR}^{\mathrm{glob}}(\mathcal{G}'))\right) + 6\log 3 + 4\epsilon\right) \\
& \underset{(c)}{\leq} \left(\sqrt{5} + \sqrt{\frac{10}{\pi}}\frac{1}{\sigma}\right)(2\|\mathcal{G} - \mathcal{G}'\|_\infty + 6\log 3 + 4\epsilon) \\
& \leq \left(\sqrt{5} + \sqrt{\frac{10}{\pi}}\frac{1}{\sigma}\right)(2\|\mathcal{G} - \mathcal{G}'\|_1 + 6\log 3 + 4\epsilon) \\
& = \mathcal{O}(\delta + \epsilon).
\end{aligned}
$$

Inequality (a) is due to the stability theorem of persistence images with Gaussian kernels (Adams et al., 2017, Theorem 10).

Inequality (b) is due to the 3-approximation theorem of Vietoris-Rips complex with Witness complex (Theorem 2.3., Arafat et al. (2020)).

Inequality (c) is due to the stability theorem of persistence diagram of Vietoris-Rips complex (Chazal et al., 2009). □

**Proposition 2** (Stability of the attention-driven node representation in Figure 1). *If the landmarks selected for the Witness complex induce an $\epsilon$-net of the graph with $\epsilon > 0$, we obtain that for any graph perturbation $\|\mathcal{G} - \mathcal{G}'\|_1 = \mathcal{O}(\delta)$ the global persistence image representation changes by*

$$
\|\boldsymbol{Z}_{\mathrm{WGTL}}(\mathcal{G}) - \boldsymbol{Z}_{\mathrm{WGTL}}(\mathcal{G}')\|_1 = \mathcal{O}((C_\epsilon + L_{GNN})(\delta + \epsilon)^2). \tag{5}
$$

*Proof. Step 1: Decomposition to three components.*

We observe that

$$
\boldsymbol{Z}_{\mathrm{WGTL}}(\mathcal{G}) = \left(\alpha_{T_L}\boldsymbol{A}(\mathcal{G}) + \alpha_G \boldsymbol{A}^{\mathrm{GNN}}(\mathcal{G})\right)\boldsymbol{A}^{glob}(\mathcal{G}),
$$

where $\boldsymbol{A}(\mathcal{G})$ represents the adjacency matrix constructed from the local topology encoding $\boldsymbol{Z}_{T_L}$ of the nodes, $\boldsymbol{A}^{glob}(\mathcal{G})$ represents the adjacency matrix constructed from the global topology encoding $\boldsymbol{Z}_{T_G}$ of the nodes, $\boldsymbol{A}^{\mathrm{GCN}}$ represents the adjacency matrix constructed from the encoding $\boldsymbol{Z}_G^{(m+1)}$ of the nodes obtained from GNNs, and $\alpha_{T_L}$ and $\alpha_G$ are non-negative attention weights in $(0, 1)$ as described in Section 4.1.

*Step 2: Stability of the three individual components.*

1. For the local PIs passing through transformer, we have

$$
\begin{aligned}
\|A(\mathcal{G}) - A(\mathcal{G}')\|_1 & \leq \sum_{l=1}^{|\mathfrak{L}|} |\mathcal{G}_l^\epsilon| \times \|\mathrm{PI}(\mathrm{Wit}(\mathcal{G}_l^\epsilon)) - \mathrm{PI}(\mathrm{Wit}(\mathcal{G}'_l^\epsilon))\|_1 \\
& \leq \left(\sqrt{5} + \sqrt{\frac{10}{\pi}}\frac{1}{\sigma}\right) C_\epsilon\left(\|\mathcal{G} - \mathcal{G}'\|_1 + 6\log 3 + 4\epsilon\right).
\end{aligned}
$$

The result follows the arguments in Theorem 1.

2. For the global PIs passing through CNN, we have

$$
\begin{aligned}
\|\boldsymbol{A}^{glob}(\mathcal{G}) - \boldsymbol{A}^{glob}(\mathcal{G}')\|_\infty &\leq \|\boldsymbol{A}^{glob}(\mathcal{G}) - \boldsymbol{A}^{glob}(\mathcal{G}')\|_1 \\
&\leq \|\mathrm{PI}^{\mathrm{glob}}(\mathcal{G}) - \mathrm{PI}^{\mathrm{glob}}(\mathcal{G}')\|_1 \\
&\leq \left(\sqrt{5} + \sqrt{\frac{10}{\pi}}\frac{1}{\sigma}\right)\left(\|\mathcal{G} - \mathcal{G}'\|_1 + 6\log 3 + 4\epsilon\right).
\end{aligned}
$$

The result follows the arguments in Proposition 1.

3. For the graph passing through GNN, following Jia et al. (2023), we show that

$$
\|A^{\mathrm{GNN}}(\mathcal{G}) - A^{\mathrm{GNN}}(\mathcal{G}')\|_1 \leq L_{\mathrm{GNN}}\|\mathcal{G} - \mathcal{G}'\|_1.
$$

*Step 3: Merging the pieces together.*

$$
\begin{aligned}
&\|\boldsymbol{Z}_{\mathrm{WGTL}}(\mathcal{G}) - \boldsymbol{Z}_{\mathrm{WGTL}}(\mathcal{G}')\|_1 \\
&= \|\left(\alpha_{T_L}\boldsymbol{A}(\mathcal{G}) + \alpha_G\boldsymbol{A}^{\mathrm{GNN}}(\mathcal{G})\right)\boldsymbol{A}^{glob}(\mathcal{G}) - \left(\alpha_{T_L}\boldsymbol{A}(\mathcal{G}') + \alpha_G\boldsymbol{A}^{\mathrm{GNN}}(\mathcal{G}')\right)\boldsymbol{A}^{glob}(\mathcal{G}')\|_1 \\
&\underset{(a)}{\leq} \|\left(\alpha_{T_L}\boldsymbol{A}(\mathcal{G}) + \alpha_G A^{\mathrm{GNN}}\right) - \left(\alpha_{T_L}\boldsymbol{A}(\mathcal{G}') + \alpha_G\boldsymbol{A}^{\mathrm{GNN}}(\mathcal{G}')\right)\|_1\|A^{glob}(\mathcal{G}) - \boldsymbol{A}^{glob}(\mathcal{G}')\|_\infty \\
&\underset{(b)}{\leq} \left(\|\boldsymbol{A}(\mathcal{G}) - A(\mathcal{G}')\|_1 + \|\boldsymbol{A}^{\mathrm{GNN}} - \boldsymbol{A}^{\mathrm{GNN}}(\mathcal{G}')\|_1\right)\|A^{glob}(\mathcal{G}) - \boldsymbol{A}^{glob}(\mathcal{G}')\|_\infty \\
&= \mathcal{O}\left(C_\epsilon(\delta + \epsilon) + L_{\mathrm{GNN}}\delta)C_\epsilon(\delta + \epsilon)\right)
\end{aligned}
$$

Step (a) is due to Hölder's inequality and Step (b) is due to triangle inequality.

The final result is due to the results of Step 2. □

### D.4 STABILITY OF THE TOPOLOGICAL LOSS

**Theorem 3** (Boundedness of Topological Loss). *Let us define*

$$
L_{topo,k}(\mathrm{T}(\mathcal{G})) \triangleq \sum_{i=1}^m (d_i - b_i)^p\left(\frac{d_i + b_i}{2}\right)^q. \tag{6}
$$

*Let us assume that the cardinality of the $\epsilon$-neighborhood of any node in $\mathcal{G}$ grows polynomially, i.e. $C_\epsilon = \mathcal{O}(\epsilon^{-M})$ for an $M > 0$. If $m$ is the number of points in the persistence diagram, $k = \max\{p, q\}$ and $2k > M$, $L_{topo,k}(\mathrm{T}(\mathcal{G}))$ is bounded, such that*

$$
L_{topo,k}(\mathrm{T}(\mathcal{G})) \leq \left(\sqrt{5} + \sqrt{\frac{10}{\pi}}\frac{1}{\sigma}\right)^{2k}\epsilon^{-2kM}\mathrm{Diam}(A(\mathcal{G})) + 2^{k-2}m\mathrm{Diam}(\mathcal{G})^{2k}. \tag{7}
$$

*Proof.*

$$
\begin{aligned}
L_{topo,k}(\mathrm{T}(\mathcal{G})) &= \sum_{i=1}^m (d_i - b_i)^p\left(\frac{d_i + b_i}{2}\right)^q \\
&\underset{(a)}{\leq} \sum_{i=1}^m \frac{(d_i - b_i)^{2p}}{2} + \frac{1}{2}\left(\frac{d_i + b_i}{2}\right)^{2q} \\
&\underset{(b)}{\leq} \sum_{i=1}^m \frac{(d_i - b_i)^{2p}}{2} + 2^{q-1}\left(b_i^{2q} + \left(\frac{d_i - b_i}{2}\right)^{2q}\right) \\
&\leq \sum_{i=1}^m \left(1 + \frac{1}{2^q}\right)\frac{(d_i - b_i)^{\max\{2p, 2q\}}}{2} + 2^{q-1}\sum_{i=1}^m b_i^{2q} \\
&= \frac{1}{2}\left(1 + \frac{1}{2^q}\right)\sum_{i=1}^m (d_i - b_i)^{\max\{2p, 2q\}} + 2^{q-1}\sum_{i=1}^m b_i^{2q}
\end{aligned}
$$

$$\leq \sum_{i=1}^{m}(d_i - b_i)^{2k} + 2^{q-1}\sum_{i=1}^{m}b_i^{2q}$$

$$\underset{(c)}{\leq} C_{A(\mathcal{G})}\mathrm{Lip}(A(\mathcal{G}))^{2k} + 2^{k-1}m\mathrm{Diam}(\mathcal{G})^{2k}$$

$$\underset{(d)}{\leq} \left(\sqrt{5} + \sqrt{\frac{10}{\pi}}\frac{1}{\sigma}\right)^{2k} C_{\epsilon}^{2k}\mathrm{Diam}(A(\mathcal{G})) + 2^{k-1}m\mathrm{Diam}(\mathcal{G})^{2k}$$

$$\underset{(e)}{=} \mathcal{O}\left(\left(\sqrt{5} + \sqrt{\frac{10}{\pi}}\frac{1}{\sigma}\right)^{2k} \epsilon^{-2kM}\mathrm{Diam}(A(\mathcal{G})) + 2^{k-1}m\mathrm{Diam}(\mathcal{G})^{2k}\right)$$

Step (a) is due to the fact that $2xy \leq x^2 + y^2$ for all $x, y \in \mathbb{R}$.

Step (b) holds true as $\left(\frac{x+y}{2}\right)^{2q} = \left(\frac{x-y}{2} + y\right)^{2q} \leq 2^{q-1}\left(\left(\frac{x-y}{2}\right)^q + y^q\right)^2 \leq 2^q \left(\left(\frac{x-y}{2}\right)^{2q} + y^{2q}\right)$ for $x, y \geq 0$ and $q \geq 1$.

Inequality (c) holds due to two facts:

i. degree $2k$-total persistence for any Lipschitz function $f$ over a triangulable compact metric space $Dom$ is upper bounded by $C_{Dom}\mathrm{Lip}(f)^{2k}$ for $2k > M$ (Cohen-Steiner et al., 2010), and

ii. the birth and death of topological features on a graph $\mathcal{G}$ is upper bounded by the diameter of the graph $\mathrm{Diam}(\mathcal{G})$, as mentioned in Section 3.

Inequality (d) is due to the Lipschitzness property of persistence images (Adams et al., 2017, Theorem 4) applied on the local persistence images calculated on an $\epsilon$-neighborhood. Thus, $\mathrm{Lip}(\boldsymbol{A}(\mathcal{G})) \leq \left(\sqrt{5} + \sqrt{\frac{10}{\pi}}\frac{1}{\sigma}\right) C_{\epsilon}$, where $C_{\epsilon}$ is the maximum cardinality of the $\epsilon$-neighborhoods induced by the landmarks.

The last inequality (e) holds due to the assumption that the maximum cardinality of local subgraph with $\epsilon$ diameter is $\mathcal{O}(\epsilon^{-M})$ (Cohen-Steiner et al., 2010). □

**Proposition 3** (Stability of Topological Loss). *Let us define*

$$L_{topo,k}(\mathrm{T}(\mathcal{G})) \triangleq \sum_{i=1}^{m}(d_i - b_i)^p \left(\frac{d_i + b_i}{2}\right)^q.$$

*Let us assume that the cardinality of the $\epsilon$-neighborhood of any node in $\mathcal{G}$ grows polynomially, i.e. $C_{\epsilon} = \mathcal{O}(\epsilon^{-M})$ for an $M > 0$. If $m$ is the number of points in the persistence diagram, $k = \max\{p, q\}$ and $2k > M$, $L_{topo,k}(\mathrm{T}(\mathcal{G}))$ is bounded, such that*

$$\left|L_{topo,k}(\mathrm{T}(\mathcal{G})) - L_{topo,k}(\mathrm{T}(\mathcal{G}'))\right|$$

$$=\mathcal{O}\left(k\epsilon^{-2k}\left(\left(\sqrt{5} + \sqrt{\frac{10}{\pi}}\frac{1}{\sigma}\right)^{2k} \epsilon^{-2kM}\mathrm{Diam}(\boldsymbol{A}(\mathcal{G})) + 2^{k-1}m\mathrm{Diam}(\mathcal{G})^{2k}\right)\|\mathcal{G} - \mathcal{G}'\|_1\right) \quad (8)$$

*Proof.*

$$|L_{topo,k}(\mathrm{T}(\mathcal{G})) - L_{topo,k}(\mathrm{T}(\mathcal{G}'))|$$

$$= |\sum_{i=1}^{m}(d_i - b_i)^p \left(\frac{d_i + b_i}{2}\right)^q - \sum_{i=1}^{m}(d_i' - b_i')^p \left(\frac{d_i' + b_i'}{2}\right)^q|$$

$$\leq 2k\left(\left(\sqrt{5} + \sqrt{\frac{10}{\pi}}\frac{1}{\sigma}\right)^{2k} \epsilon^{-2kM}\mathrm{Diam}(\boldsymbol{A}(\mathcal{G})) + 2^{k-1}m\mathrm{Diam}(\mathcal{G})^{2k}\right)\|\boldsymbol{A}(\mathcal{G}) - \boldsymbol{A}(\mathcal{G}')\|_{\infty}$$

$$= \mathcal{O}\left(kC_\epsilon\left(\left(\sqrt{5} + \sqrt{\frac{10}{\pi}}\frac{1}{\sigma}\right)^{2k} \epsilon^{-2kM} \mathrm{Diam}(\boldsymbol{A}(\mathcal{G})) + 2^{k-1}m\mathrm{Diam}(\mathcal{G})^{2k}\right)\|\mathcal{G} - \mathcal{G}'\|_1\right)$$

The first inequality is a direct consequence of Theorem 3. The second inequality is due to the $\epsilon$-neighbourhood based construction of the topological loss from $\mathcal{G}$ (refer to Figure 2 and proof of Theorem 1). $\qquad\square$

# E    LANDMARK SELECTION ALGORITHM

The pseudocode for selecting landmarks for computing global witness topological features and local witness topological features is presented in Algorithm 1. In order to compute global witness features, we select a set of *global landmark* nodes. In order to compute local witness features, we select a set of *local landmark* nodes for each node in the graph.

---

**Algorithm 1 Greedy** Landmark selection algorithm

---

**Require:**  Graph $\mathcal{G} = (\mathcal{V}, \mathcal{E})$, percentage of nodes as landmarks $p \in (0, 1)$
**Ensure:**  Global landmark set $\mathfrak{L}_g$ and Local landmark set $\mathfrak{L}$
1: Number of landmarks $n_g \leftarrow |\mathcal{V}| \cdot p$
2: Sort $\mathcal{V}$ in decreasing order of node degrees.
3: Select Global Landmarks $\mathfrak{L}_g \leftarrow \mathcal{V}[1, 2, \ldots, n_g]$
4: **for all** $l \in \mathfrak{L}_g$ **do**
5:     Compute cover $C_l \leftarrow \{u \in \mathcal{V} : d_{\mathcal{G}}(u, l) < d_{\mathcal{G}}(u, l') \quad \forall l' \in \mathfrak{L}_g \setminus \{l\}\}$
6:     Compute Subgraph $G_l \leftarrow \mathcal{G}[C_l]$
7:     Number of local landmarks $n_l \leftarrow |C_l| \cdot p$
8:     Sort $C_l$ in decreasing order of node degrees in $G_l$.
9:     Select Local landmarks $\mathfrak{L}[l] \leftarrow C_l[1, \ldots, n_l]$
    **return** $\mathfrak{L}_g, \mathfrak{L}$

---

In Lines 1-3, we select the set of global landmarks $\mathfrak{L}_g$ in order to construct Global witness filtration. We select the top-most $p\%$ highest degree nodes in the graph $\mathcal{G}$ as landmarks.

In Lines 4-9, we select local landmarks corresponding to each global landmark in order to compute the topological features local to each global landmark. A node $u$ that is not a global landmark must be in the cover of some landmark node $l \in \mathfrak{L}_g$. We say $u$ is a witness node to the node $l$. We assume the local topological signature does not change inside a cover. In other words, a witness node has the same topological signature as its associated landmark. That is why, instead of computing local landmarks for every node in $\mathcal{V}$, we compute only for the global landmarks $\mathfrak{L}_g \subseteq \mathcal{V}$ (line 4). For each global landmark $l \in \mathfrak{L}_g$, we construct its cover $C_l$ in line 5 consisting of all its witness nodes. In line 6, we construct the subgraph $\mathcal{G}[C_l]$ induced by the witness nodes. Finally, in lines 7-9, we select the topmost $p\%$ of the witness nodes with the highest degrees in the induced subgraph $\mathcal{G}[C_l]$ as the local landmark for $l \in \mathfrak{L}_g$.

## E.1    IMPACT OF THE NUMBER OF LANDMARKS

It is well-known that the quality of the Witness complex-based topological features is dependent on the number of landmarks (De Silva & Carlsson, 2004a; Arafat et al., 2019). Hence, the performance of the proposed witness topological encodings, topological loss, and, finally, the downstream classification quality is also dependent on the number of landmarks.

In order to study the accuracy and efficiency of WGTL under different numbers of landmarks, we use Algorithm 1 to select $10\%, 20\%, 30\%$ nodes as landmarks, and in Figure 3, we present the accuracy and computation time of the local and global witness complex-based features on Cora-ML and Citeseer datasets. We observe that increasing the number of landmarks indeed slightly increases the accuracy, albeit with the expense of increased computation time. Due to this trade-off between accuracy and efficiency, the selection of an optimal number of landmarks is dependent on how much robustness is desired by a user within a given computation-time budget.

## E.2    COMPARISON OF WGTL AND VIETORIS-RIPS BASED TOPOLOGY ENCODING (VRGTL)

Finally, we also observe that, on these datasets, the accuracy achieved by WGTL with 30% landmarks is close to the accuracy achieved by adopting Vietoris-Rips-based topological feature encoding, as indicated by the dotted line representing GCN+VRGTL. A more in-depth comparison among GCN, GCN+WGTL and GCN+VRGTL is presented in Figure 3 where we compare their accuracy on Cora-ML and Citeseer under mettack. We observe that the accuracy of GCN+VRGTL is comparable to that of GCN+WGTL. These observations also highlight the flexibility of WGTL in adopting other approximate topological features. However, computing Vietoris-Rips features is significantly more

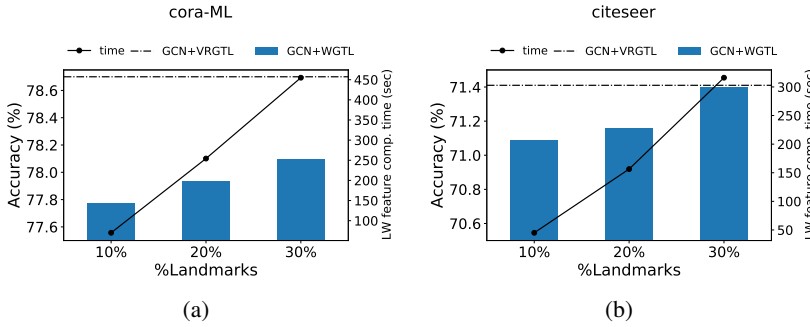

Figure 3: The trade-off between accuracy and Feature computation time of GCN+WGTL with different numbers of landmarks. The figures are under mettack with $5\%$ perturbation rate.

Table 12: Performance (Accuracy±Std) comparison with GCN+VRGTL under mettack. GCN+VRGTL has a similar performance as GCN+WGTL as their mean±std coincide.

| Dataset | Model | Perturbation Rate | | | | | |
|---|---|---|---|---|---|---|---|
| | | 0% | 5% | 10% | 15% | 20% | 25% |
| Cora-ML | GCN | 82.87±0.83 | 76.55±0.79 | 70.39±1.28 | 65.10±0.71 | 52.30±1.43 | 47.53±1.96 |
| | GCN + VRGTL (ours) | *84.02 ± 0.5 | *78.70 ±0.65 | 73.07 ±0.74 | 67.77±0.52 | 56.93±0.58 | 52.39±0.77 |
| | GCN + WGTL (ours) | 83.83±0.55 | 78.63±0.76 | *73.41±0.82 | *68.87±0.89 | *57.47±1.00 | *53.71±1.81 |
| Citeseer | GCN | 71.56±0.56 | 67.28±0.61 | 62.49±0.88 | 55.70±1.2 | 49.23±1.06 | 49.00±1.67 |
| | GCN + VRGTL (ours) | *72.64 ± 0.6 | *71.41 ±0.45 | 67.93 ±0.41 | 65.84±0.87 | *57.00±0.80 | *58.00 ± 0.72 |
| | GCN + WGTL (ours) | 72.56±0.82 | 71.40±0.93 | *67.95±0.43 | *65.97±0.40 | 55.84±1.44 | 57.95±0.70 |

Table 13: Execution times for computing global and local topological features. Landmark selection time is included.

| | Cora-ML | Citeseer |
|---|---|---|
| VR feature comp. time (seconds) | 23286 | 1698 |
| Witness feature comp. time (seconds) | 24 | 16.2 |

expensive than witness topological features (Arafat et al., 2020). As shown in Table 13, adopting VRGTL on Cora-ML (Citeseer) incurs almost 1000x (100x) more computation time compared to WGTL.

*The results demonstrate that instead of incurring 2 to 3 order less computational time, deploying WGTL leads to similar or better accuracy to those of VRGTL across a wide range of perturbations.*

# F  ADDITIONAL EXPERIMENTAL RESULTS

## F.1  WGTL WITH DIFFERENT GNN BACKBONE ARCHITECTURES

In the main paper, we have deployed WGTL with two GNN backbone architectures: GCN Kipf & Welling (2016) and Pro-GNN Jin et al. (2020). In order to test the versatility and flexibility of WGTL, we adopt WGTL with two more recent GNN architectures, namely GAT (Veličković et al., 2017) and GraphSAGE (Hamilton et al., 2017) as backbones. We adopt the experimental setup and landmark selection scheme described earlier in Appendix A. For the attacks, we adopt the configurations and budgets as described earlier in Appendix B.

We demonstrate the performance of the backbones with and without WGTL on two representative datasets: Cora-ML and Polblogs, and under two different representative attacks, i.e., mettack and nettack. *We observe that incorporating WGTL into all of the backbones improve their corresponding performances under a range of perturbation rates.*

**GAT backbone.** Table 14 shows the performance of GAT and the proposed GAT+WGTL on the Cora-ML and Polblogs datasets under mettack. We observe that GAT+WGTL improves robustness with respect to the baseline GAT by 0.1% - 13.8% on Cora-ML and 0.6% - 50.2% on Polblogs. Table 15 shows the performance under nettack. We observe that GAT+WGTL improves robustness with respect to the baseline GAT by 0.7% - 1.1% on Cora-ML and 0.14% - 1.5% on Polblogs.

**GraphSAGE backbone.** Table 16 shows the performance of GraphSAGE and the proposed Graph-SAGE+WGTL on the Cora-ML and Polblogs datasets under mettack. We observe that Graph-SAGE+WGTL improves robustness with respect to the baseline GraphSAGE by 3.2% - 42.6% on Cora-ML and 1.5% - 40.8% on Polblogs. Table 17 shows the performance under nettack. We observe that GraphSAGE+WGTL improves robustness with respect to the baseline GraphSAGE by 3% - 4% on Cora-ML and 0.4% - 1.7% on Polblogs.

Table 14: Performance (Accuracy±Std) of GAT and GAT+WGTL under mettack.

| Dataset | Model | Perturbation Rate | | | | | |
|---|---|---|---|---|---|---|---|
| | | 0% | 5% | 10% | 15% | 20% | 25% |
| Cora-ML | GAT | 84.25±0.67 | 79.88±1.09 | 72.63±1.56 | 68.12±1.81 | 56.49±2.60 | 51.15±1.63 |
| | GAT + WGTL (ours) | **86.07±2.10** | **80.80±0.87** | **75.80±0.79** | **69.86±1.77** | **64.29±1.59** | **51.21±1.19** |
| Polblogs | GAT | 95.28±0.51 | 75.83±0.90 | 73.11±1.20 | 68.98±1.48 | 53.21±12.13 | 46.48±9.09 |
| | GAT + WGTL (ours) | **95.87±0.26** | **83.13±0.32** | **80.06±0.50** | **75.05±0.68** | **74.03±1.06** | **69.83±0.77** |

Table 15: Performance (Accuracy±Std) of GAT and GAT+WGTL under nettack.

| Dataset | Model | Perturbation Rate | | | | | |
|---|---|---|---|---|---|---|---|
| | | 0% | 5% | 10% | 15% | 20% | 25% |
| Cora-ML | GAT | 84.25±0.67 | 83.92±0.65 | 83.11± 0.42 | 82.94±0.59 | 82.34±0.55 | 81.72±0.56 |
| | GAT + WGTL (ours) | **86.07±2.10** | **84.53±0.75** | **83.72±0.61** | **83.82±0.73** | **83.23±0.52** | **82.31±0.30** |
| Polblogs | GAT | 95.28±0.51 | 89.86±0.63 | 86.44±1.47 | 86.40±1.47 | 86.28±2.72 | 85.15±2.81 |
| | GAT + WGTL (ours) | **95.87±0.26** | **90.69±0.51** | **87.73±0.38** | **87.22±0.31** | **86.63±1.12** | **85.27±0.29** |

## F.2  COMPARISON WITH EXISTING DEFENSE MECHANISMS: GNNGUARD

Tables 18 shows the performance comparison between (1) GCN, (2) GCN+GNNGuard (Zhang & Zitnik, 2020), (3) GCN+WGTL, and (4) GCN+GNNGuard+WGTL on Cora-ML and Polblogs datasets under mettack. We can observe that (i) as expected, GCN+GNNGuard+WGTL always outperforms all baselines, e.g., on Cora-ML, GCN+GNNGuard+WGTL yields more than 2.3%, 1.1%, and 1.9% relative improvements to GCN, GCN+GNNGuard, and GCN+WGTL respectively, (ii) in general, GCN+WGTL is better than GCN+GNNGuard on both Cora-ML and Polblogs

Table 16: Performance (Accuracy±Std) of GraphSAGE and GraphSAGE+WGTL under mettack.

| Dataset | Model | Perturbation Rate | | | | | |
|---------|-------|-----|-----|-----|-----|-----|-----|
| | | 0% | 5% | 10% | 15% | 20% | 25% |
| Cora-ML | GraphSAGE | 81.00±0.27 | 74.81±1.2 | 70.92±1.18 | 67.46±0.80 | 60.54± 2.08 | 53.48±0.74 |
| | GraphSAGE + WGTL (ours) | **83.63±0.35** | **82.61±0.65** | **81.19±1.13** | **80.06±0.18** | **78.10±1.07** | **76.28±0.31** |
| Polblogs | GraphSAGE | 94.52±0.27 | 77.44 ± 1.71 | 74.66±0.85 | 68.77±1.83 | 59.65±1.77 | 54.15±2.10 |
| | GraphSAGE + WGTL (ours) | **95.58±0.50** | **82.62±0.65** | **81.49±0.86** | **80.06±1.21** | **76.79±0.54** | **76.27±0.22** |

Table 17: Performance (Accuracy±Std) of GraphSAGE and GraphSAGE+WGTL under nettack.

| Dataset | Model | Perturbation Rate | | | | | |
|---------|-------|-----|-----|-----|-----|-----|-----|
| | | 0% | 5% | 10% | 15% | 20% | 25% |
| Cora-ML | GraphSAGE | 81.01±0.27 | 80.48±0.71 | 80.19±0.49 | 79.85±0.87 | 78.72±0.32 | 78.41±1.19 |
| | GraphSAGE + WGTL (ours) | **83.63±0.35** | **83.23±0.21** | **82.79±0.36** | **82.23±0.61** | **81.88±0.40** | **81.49±0.59** |
| Polblogs | GraphSAGE | 94.54±0.27 | 90.20±0.30 | 89.57±0.62 | 89.28±0.82 | 88.30±1.0 | 87.06±2.15 |
| | GraphSAGE + WGTL (ours) | **95.58±0.50** | **90.98±0.27** | **89.95±0.78** | **89.85±0.98** | **88.92±1.13** | **88.56±0.36** |

Table 18: Performance (Accuracy±Std) of WGTL, GNNGuard, and GNNGuard+WGTL with GCN backbone under Mettack.

| Dataset | Model | Perturbation Rate | | | | | |
|---------|-------|-----|-----|-----|-----|-----|-----|
| | | 0% | 5% | 10% | 15% | 20% | 25% |
| Cora-ML | GCN | 82.87±0.83 | 76.55±0.79 | 70.39±1.28 | 65.10±0.71 | 52.30±1.43 | 47.53±1.96 |
| | GCN + WGTL (ours) | 83.83±0.55 | 78.63±0.76 | 73.41±0.82 | 68.87±0.89 | 57.47±1.00 | 53.71±1.81 |
| | GCN + GNNGuard | 83.21±0.34 | 76.57±0.50 | 69.13±0.77 | 65.29±0.84 | 55.85±0.67 | 51.51±1.0 |
| | GCN + GNNGuard + WGTL (ours) | **84.78±0.43** | **83.23±0.82** | **79.96±0.49** | **79.90±0.94** | **76.07±0.89** | **75.35±0.78** |
| Polblogs | GCN | 94.40±1.47 | 71.41±2.42 | 69.16±1.86 | 64.66±2.59 | 56.05±2.18 | 48.59±1.44 |
| | GCN + WGTL (ours) | 95.95±0.15 | 74.62±0.42 | 72.84±0.86 | 68.65±0.31 | 62.44±1.51 | 58.24±0.14 |
| | GCN + GNNGuard | 95.03±0.25 | 73.25±0.16 | 72.76±0.75 | 69.18±0.20 | 61.57±0.65 | 57.14±0.82 |
| | GCN + GNNGuard + WGTL (ours) | **96.22±0.25** | **75.25±0.81** | **73.04±1.46** | **70.14±0.62** | **63.19±0.76** | **61.60±0.81** |

Table 19: Performance (Accuracy±Std) of WGTL, GNNGuard, and GNNGuard+WGTL with GCN backbone under nettack.

| Dataset | Model | Perturbation Rate | | | | | |
|---------|-------|-----|-----|-----|-----|-----|-----|
| | | 0 | 1 | 2 | 3 | 4 | 5 |
| Cora-ML | GCN | 82.87±0.93 | 82.53±1.06 | 82.08±0.81 | 81.69 ±0.59 | 81.26±0.88 | 80.69±0.81 |
| | GCN + WGTL (ours) | 83.83±0.55 | 83.41±0.87 | 82.74±0.65 | 82.06±0.82 | 81.64±0.55 | 80.98±0.67 |
| | GCN + GNNGuard | 83.21 ±0.34 | 82.81±0.43 | 82.51±0.26 | 82.03±0.30 | 81.61±0.25 | 80.79±0.26 |
| | GCN + GNNGuard + WGTL (ours) | **84.78±0.43** | **84.25±0.73** | **83.74±0.96** | **83.70±1.01** | **84.05±0.60** | **83.84±0.26** |
| Polblogs | GCN | 94.40±1.48 | 88.91±1.06 | 85.39±0.86 | 83.03±0.87 | 81.20±1.63 | 79.39±0.96 |
| | GCN + WGTL (ours) | 95.95±0.15 | 91.47±0.33 | 89.10±0.69 | 88.98±0.83 | 88.63±1.20 | 87.14±0.70 |
| | GCN + GNNGuard | 95.03±0.25 | 91.43±0.36 | 89.45±0.46 | 90.04±0.06 | 89.16±0.36 | 88.94±0.72 |
| | GCN + GNNGuard + WGTL (ours) | **96.22±0.25** | **91.89±0.57** | **90.35±0.81** | **90.13±1.92** | **89.86±0.83** | **89.19±0.57** |

datasets. Table 19 displays the performance comparison between (1) GCN, (2) GCN+GNNGuard, (3) GCN+WGTL, and (4) GCN+GNNGuard+WGTL on Cora-ML and Polblogs datasets under nettack. Similarly, the results of GCN +GNNGuard+WGTL are consistently better than all the other baselines.

Table 20: Performance of the algorithms on a heterophilic graph: snap-patents under Mettack. We compare $H_2$GCN (Zhu et al., 2020) and $H_2$GCN+WGTL for heterophilic graphs.

| Dataset | Models | Perturbation Rate | | | | | |
|---|---|---|---|---|---|---|---|
| | | 0% | 5% | 10% | 15% | 20% | 25% |
| snap-patents | $H_2$GCN | 27.71±0.86 | 27.55±0.19 | 28.62±0.38 | 28.40±1.38 | 27.77±0.30 | 27.45±0.89 |
| | $H_2$GCN+WGTL | **27.72±0.85** | **28.66±1.68** | **28.79±1.0** | **28.45±0.61** | **28.21 ±0.66** | **27.90±0.84** |

## F.3 WGTL on Heterophilic Graphs

In the previous experiments, we have used four homophilic graph datasets: Cora-ML, Citeseer, Pubmed, and Polblogs. In this section, we aim to test the performance of WGTL on a heterophilic graph. Adopting the same attack configurations described in Appendix B, we generate different perturbations (perturbation rates 0% to 25%) of snap-patents graph (Leskovec et al., 2005).

**Heterophilic Graph Dataset.** The snap-patents is a utility patent citation network. Node labels reflect the time the patent was granted, and the features are derived from the patent's metadata. Following Zhu et al. (2022), for better computational tractability, we sample a subset of the snap-patents data using a snowball sampling approach, where a random 20% of the neighbors for each traversed node are kept. The snap-patents graph contains 4562 nodes, 12103 edges, 5 classes, and 269-dimensional node features. The homophily ratio for this dataset is as low as $0.134$. Hence, Zhu et al. (2022) used this dataset as a benchmark heterophilic graph to test the robustness of many GNN architectures, including those proposed for heterophilic graphs.

**Backbone GNN: $H_2$GCN.** Recently, it has been shown that GCN and other classical GNNs (e.g. GAT) perform poorly on heterophilic graphs (Zhu et al., 2020; 2021). More recently, Zhu et al. (2022) showed that these classical methods provide poor defence against adversarial attacks on heterophilic graphs as well. Thus, instead of GCN, we adopt $H_2$GCN (Zhu et al., 2020) as the backbone architecture in this experiment. $H_2$GCN is proposed and popularly deployed to conduct classification on the heterophilic graphs. $H_2$GCN proposes a set of key design techniques to improve performance of GNNs on heterophilic graphs: (1) separation of ego- and neighbor-embedding, (2) incorporation of higher-order neighborhoods, and (3) combination of intermediate representations using skip-connections.

**Results and Observations.** For each perturbation rate, we run five experiments with $H_2$GCN and $H_2$GCN+WGTL on the corresponding perturbed snap-patents dataset, and report the mean± standard deviation of the final classification accuracy in Table 20. The results show that $H_2$GCN+WGTL robustly improves the accuracy over $H_2$GCN by up to 4% across the perturbation rates. Note that the best-performing method (APPNP (Gasteiger et al., 2018)) on this dataset has been shown to have an accuracy of 27.76% under 20% perturbation (c.f. Table 3, Zhu et al. (2022). Improving on that, we observe that $H_2$GCN+WGTL achieves 28.21% average accuracy under 20% perturbation.

