\|A^{glob}(\mathcal{G}) - A^{glob}(\mathcal{G}')\|_\infty &\le \|A^{glob}(\mathcal{G}) - A^{glob}(\mathcal{G}')\|_1 \\
&\le \|\mathrm{PI}^{\mathrm{glob}}(\mathcal{G}) - \mathrm{PI}^{\mathrm{glob}}(\mathcal{G}')\|_1 \\
&\le \left(\sqrt{5} + \sqrt{\frac{10}{\pi}}\frac{1}{\sigma}\right)(\|\mathcal{G} - \mathcal{G}'\|_1 + 6\log 3 + 4\epsilon).
\end{aligned}
$$

The result follows the arguments in Proposition 1.

3. For the graph passing through GNN, following Jia et al. (2023), we show that

$$
\|A^{\mathrm{GNN}}(\mathcal{G}) - A^{\mathrm{GNN}}(\mathcal{G}')\|_1 \le L_{\mathrm{GNN}}\|\mathcal{G} - \mathcal{G}'\|_1.
$$

*Step 3: Merging the pieces together.*

$$
\begin{aligned}
&\|\boldsymbol{Z}_{\mathrm{WGTL}}(\mathcal{G}) - \boldsymbol{Z}_{\mathrm{WGTL}}(\mathcal{G}')\|_1 \\
&= \|\left(\alpha_{T_L}A(\mathcal{G}) + \alpha_G A^{\mathrm{GNN}}\right)A^{glob}(\mathcal{G}) - \left(\alpha_{T_L}A(\mathcal{G}') + \alpha_G A^{\mathrm{GNN}}(\mathcal{G}')\right)A^{glob}(\mathcal{G}')\|_1 \\
&\underset{(a)}{\le} \|\left(\alpha_{T_L}A(\mathcal{G}) + \alpha_G A^{\mathrm{GNN}}\right) - \left(\alpha_{T_L}A(\mathcal{G}') + \alpha_G A^{\mathrm{GNN}}(\mathcal{G}')\right)\|_1 \|A^{glob}(\mathcal{G}) - A^{glob}(\mathcal{G}')\|_\infty \\
&\underset{(b)}{\le} \left(\|A(\mathcal{G}) - A(\mathcal{G}')\|_1 + \|A^{\mathrm{GNN}} -