# OpenReview forum: "When Witnesses Defend: A Witness Graph Topological Layer for Adversarial Graph Learning"
_ICLR.cc/2024/Conference — Submitted to ICLR 2024_

### Official Review · Reviewer_EvXo · 2023-10-30

**Soundness:** 3 good
**Presentation:** 3 good
**Contribution:** 3 good
**Rating:** 5
**Confidence:** 2

**Summary:**

This work designs Witness Graph Topological Layer (WGTL), which systematically integrates both local and global topological graph feature representations whose impact are in turn automatically controlled by the robust regularized topological loss. Some experiments are conduct to show the effectiveness of the method.

**Strengths:**

1. The paper is well-written.
2. The idea is supported by a theorectical foundation.
3. The experiments show the improvement against baselines.

**Weaknesses:**

1. The paper only compared with vanilla GCN. I believe more baselines including some SOTA defense methods should be included. Without this comparison, I lean to a weak reject.
2. The threat model should be moved to the main body of the paper, instead of in the appendix.

**Questions:**

See weakness.

---

> ### Author Response · Authors · 2023-11-20
> **Response to Technical Questions**
>
> We thank the reviewer for the feedback to improve our experimental analysis and exposition.
>
> > (1) Further comparison with different baselines
>
> In the revised draft, we demonstrate the performance of WGTL with different GNN backbones, with a backbone specific to heterophilic graphs. In addition, we also compare WGTL with another existing defense mechanism.
>
>  (a)    **Performance of WGTL with different GNN backbones:** In addition to GCN, We have added two more backbones, GAT [6] (Tables 14-15) and GraphSAGE [1] (Tables 16-17) in Appendix F and shown the effectiveness of WGTL on Cora-ML and Polblogs under mettack and nettack. We observe that incorporating WGTL into all of the backbones improves their corresponding performances under a range of perturbation rates. The following table summarizes the %improvement in mean accuracy with respect to the corresponding backbone models:
>
> $$
> \\begin{array}{|c|c|c|c|}
> \\hline
>  & & \\% \text{improvement on Cora-ML} & \\% \text{improvement on Polblogs}  \\\\
> \hline
> GAT+WGTL & Mettack & 0.1-13.8 & 0.6-50.2 \\\\
> GraphSAGE+WGTL & Mettack &       3.2-42.6   &   1.5-40.8 \\\\
> GAT+WGTL    	& Nettack &  0.7-1.1     &   0.14-1.5 \\\\
> GraphSAGE+WGTL & Nettack &    3-4   &    0.4-1.7 \\\\
> \\hline
> \\end{array}
> $$
> [(URL to Tables 14-15)](https://drive.google.com/file/d/1315msJ6yJGCoKncR2Fd6sWimv3tSVj5n/view?usp=sharing)
>
> [(URL to Tables 16-17)](https://drive.google.com/file/d/138WvQLB6ieS-fojA9D1brAKc3NnbSm1H/view?usp=sharing)
>
> (b)    **Performance of WGTL with a heterophilic graph-specific backbone (H$_2$GCN [3]):** Following [4], We experiment with snap-patents, a strongly heterophilic graph under mettack. We incorporated global topology encoding into H${_2}$GCN [3], a popular method with 580+ citations proposed to handle heterophilic graphs. In Table 20 of Appendix F3, we have shown that H${_2}$GCN+WGTL improves the robustness of H$_2$GCN by up to 4%.
> Note that the most adversarially robust method on this dataset, APPNP by [5], has been shown to have an accuracy of $27.76\%$ under 20\% perturbation (c.f. Table 3, [4]). Improving on that, we observe that H${_2}$GCN+WGTL achieves $28.21\%$ average accuracy under $20\%$ perturbation.
> $$
> \bf{Table 20 =>}
> \\begin{array}{|c|c|c|c|c|c|c|}
> \\hline
>  \text{Perturbation rates=>} & 0\\% & 5\\% & 10\\%  & 15\\% & 20\\% & 25\\% \\\\
> \hline
> H_2GCN & 27.71\pm 0.86  	& 27.55 \pm 0.19  & 28.62 \pm 0.38 & 28.40 \pm 1.38 & 27.77 \pm 0.30 & 27.45 \pm0.89  \\\\
> H_2GCN+WGTL & 27.72 \pm 0.85 & 28.66 \pm 0.1.68  & 28.79 \pm 1.0 & 28.45 \pm 0.61 & 28.71 \pm 0.66 & 27.90 \pm 0.84 \\\\
> \\hline
> \\end{array}
> $$
>
> (c)    **Comparing with more SOTA defenses for GNNs:** In Appendix F.2, we have compared WGTL with two SOTA defense methods in the paper: ProGNN (Table 5) and GNNGuard [2] (Tables 18-19). With respect to ProGNN (with GCN backbone) SOTA, our method ProGNN+WGTL gains 0.68% - 4.96% of relative improvements on Cora-ML and Citeseer under mettack. On Cora-ML under mettack, our method GCN+GNNGuard+WGTL yields more than 2.25%, 1.12%, and 1.85% relative improvements with respect to GCN, GCN+GNNGuard (SOTA defence), and GCN+WGTL, respectively.
>
> [(URL to Table 5)](https://drive.google.com/file/d/13AJoUfgaXf_XHYylovvJVWG97YsoV4u2/view?usp=sharing)
>
> [(URL to Tables 18-19)](https://drive.google.com/file/d/13942jxN8ZntEusr3SgL93LScd2G_PRJ7/view?usp=sharing)
>
> > (2) Positioning the attacks in the main paper
>
> Thank you for this suggestion. Due to page limitations, we are unable to move Appendix B to the main body. Presently, we provide a brief description of different attacks considered in the experiments (Section 5) in the paragraph “Adversarial Attacks: Local and Global.” (page 7). However, if the reviewer thinks it to be critical, we will try our best to do it.
>
> References:
>
> [1]  Will Hamilton, Zhitao Ying, and Jure Leskovec. Inductive representation learning on large graphs.
> NeurIPS, 30, 2017.
>
> [2] Xiang Zhang and Marinka Zitnik. Gnnguard: Defending graph neural networks against adversarial
> attacks. NeurIPS, 33:9263–9275, 2020
>
> [3] Jiong Zhu, Yujun Yan, Lingxiao Zhao, Mark Heimann, Leman Akoglu, and Danai Koutra. Beyond
> homophily in graph neural networks: Current limitations and effective designs. NeurIPS, 33:7793–7804, 2020
>
> [4] Jiong Zhu, Junchen Jin, Donald Loveland, Michael T Schaub, and Danai Koutra. How does
> heterophily impact the robustness of graph neural networks? theoretical connections and practical
> implications. In Proceedings of the 28th ACM SIGKDD Conference on Knowledge Discovery and
> Data Mining, pp. 2637–2647, 2022
>
> [5] Johannes Gasteiger, Aleksandar Bojchevski, and Stephan Gunnemann. Predict then propagate: Graph
> neural networks meet personalized pagerank. arXiv preprint arXiv:1810.05997, 2018
>
> [6] Petar Velickovic, Guillem Cucurull, Arantxa Casanova, Adriana Romero, Pietro Lio, and Yoshua
> Bengio. Graph attention networks. arXiv preprint arXiv:1710.10903, 2017.

---

> > ### Comment · Reviewer_EvXo · 2023-11-21
> > **Thank you for your rebuttal.**
> >
> > I appreciate the efforts in the rebuttal and the rebuttal partially solves my concerns.

---

> > > ### Author Response · Authors · 2023-11-22
> > > **Response to Reviewer EvXo**
> > >
> > > Dear Reviewer EvXo,
> > >
> > > Thank you again for the time and effort that you dedicated to providing invaluable feedback on our paper. We would very much appreciate if you could consider updating the scores before the deadline (rapidly approaching). Please let us know If there are any remaining concerns.
> > >
> > > Best,
> > >
> > > Paper ID 5896 Authors

---

> ### Author Response · Authors · 2023-11-21
>
> Thank you for considering our response. We are glad to be able to resolve your concerns partially. If you have any further questions so as to resolve your concerns fully, please let us know. Otherwise, please kindly consider changing your score accordingly.

---

### Official Review · Reviewer_REKj · 2023-10-31

**Soundness:** 2 fair
**Presentation:** 3 good
**Contribution:** 3 good
**Rating:** 6
**Confidence:** 4

**Summary:**

The authors propose an adversarial defense strategy for graph neural networks that primarily relies on Persistent Homology (PH) representations of graphs. The key intuition behind the authors' method is to estimate the salient anatomy nodes of graph-structured data, while regarding remaining nodes as witnesses to enhance the robustness of node representations. Building on these concepts, this paper introduces the Witness Graph Topological Layer (WGTL), which takes into account both local and global topological graph features to improve model robustness. The authors validate their method against global and local poisoning attacks on citation graphs using the GCN architecture and demonstrate that their topological layer and regularized topological loss can enhance the robustness of GCN.

**Strengths:**

- The proposed strategy is intuitive and somewhat straightforward for enhancing robustness on graph neural networks.
- The authors also provide theoretical analysis regarding stability for topology encodings and regularized topological loss against perturbations.
- Overall, the draft is well-written and easy to follow, although there is room for improvement in section organization and figures.
- The improvements in robustness capability appear to be significant when compared to the vanilla GNN.

**Weaknesses:**

I have major concerns, mostly regarding the experimental evaluation section. I find the current form of the experiments to be lacking in several aspects for the following reasons:
- There is a lack of comparisons with previous non-topological works. While I agree that this work would be the first to utilize persistent homology, previous baseline methods also consider similar information/knowledge, such as neighboring structure or connectivity patterns using other concepts. Simply demonstrating improvements over the vanilla GCN or combining with a single specific baseline may not be entirely convincing.
- The entire set of experiments is conducted using GCN. The significant improvement observed in GCN may be attributed to its naive mechanism. Given that the incorporation of a multi-scale receptive field for node representation is already well-explored in the field of graph neural networks, it is highly recommended to conduct experiments on more recent graph neural networks.
- Can this method also perform well on heterophilous graphs or molecular graphs? The proposed method has only been validated on homophilous graphs, particularly citation networks. Therefore, it remains uncertain whether this strategy can exhibit versatility when applied to other types of graphs.

**Questions:**

Please address the concerns mentioned in the weaknesses section. I also recommend moving the related work section to the beginning. Readers might find it challenging to follow or understand the existing approaches for tackling adversarial attacks on graph neural networks.

---

> ### Author Response · Authors · 2023-11-20
> **Response to Technical Questions (1/2)**
>
> We thank the reviewer for his detailed feedback to improve the paper and specially, for introducing us to the interesting line of works on graphs with heterophily to instantiate flexibility and generality of WGTL.
> > (1) Further comparison with different baselines
>
> Thank you for pointing out this limitation. We have compared and demonstrated improvement over non-topological works, in particular those that use neighboring structure or connectivity patterns using other concepts such as GraphSAGE, GNNGuard, and H$_2$GCN.
>
> (a)   **Performance with GraphSAGE [1]:** Table 16 shows the performance of GraphSAGE and the proposed GraphSAGE+WGTL on the Cora-ML and Polblogs datasets under mettack. We observe that GraphSAGE+WGTL improves robustness with respect to the baseline GraphSAGE by 3.2% - 42.6% on Cora-ML and 1.5% - 40.8% on Polblogs. Table 17 shows the performance under nettack. We observe that GraphSAGE+WGTL improves robustness with respect to the baseline GraphSAGE by 3% - 4% on Cora-ML and 0.4% - 1.7% on Polblogs.
>
> [(URL to Tables 16-17)](https://drive.google.com/file/d/138WvQLB6ieS-fojA9D1brAKc3NnbSm1H/view?usp=sharing)
>
> (b)    **Comparison with GNNGuard [2]:** In Appendix F.2, we have compared the proposed WGTL with GNNGuard (Tables 18-19). On Cora-ML under mettack (Table 18), our method GCN+GNNGuard+WGTL yields more than 2.25%, 1.12%, and 1.85% relative improvements with respect to GCN, GCN+GNNGuard (SOTA defence), and GCN+WGTL, respectively.
>
> [(URL to Tables 18-19)](https://drive.google.com/file/d/13942jxN8ZntEusr3SgL93LScd2G_PRJ7/view?usp=sharing)
>
> (c) **Performance with  H${_2}$GCN [3]:** Following [4], we run experiments on snap-patents, a strongly heterophilic graph under mettack. We incorporated global topology encoding into H${_2}$GCN [3], a popular method with 580+ citations proposed to handle heterophilic graphs. H${_2}$GCN proposes a set of key design techniques to improve the performance of GNNs on heterophilic graphs: (1) separation of ego- and neighbor-embedding, (2) incorporation of higher-order neighborhoods, and (3) combination of intermediate representations using skip-connections. In Table 20 of Appendix F3, we have shown that H${_2}$GCN+WGTL improves the robustness of H2GCN by up to 4%.
> Note that the most adversarially robust method on this dataset, APPNP by [5], has been shown to have an accuracy of $27.76\%$ under 20\% perturbation (c.f. Table 3, [4]). Improving on that, we observe that H${_2}$GCN+WGTL achieves $28.21\%$ average accuracy under $20\%$ perturbation.
> $$
> \bf{Table 20 =>}
> \\begin{array}{|c|c|c|c|c|c|c|}
> \\hline
>  \text{Perturbation rates=>} & 0\\% & 5\\% & 10\\%  & 15\\% & 20\\% & 25\\% \\\\
> \hline
> H_2GCN & 27.71\pm 0.86  	& 27.55 \pm 0.19  & 28.62 \pm 0.38 & 28.40 \pm 1.38 & 27.77 \pm 0.30 & 27.45 \pm0.89  \\\\
> H_2GCN+WGTL & 27.72 \pm 0.85 & 28.66 \pm 0.1.68  & 28.79 \pm 1.0 & 28.45 \pm 0.61 & 28.71 \pm 0.66 & 27.90 \pm 0.84 \\\\
> \\hline
> \\end{array}
> $$
>
> > (2) Conduct experiments on more recent graph neural networks than GCN
>
> In addition to GCN in the main paper, we have added two more backbones, GAT [6] (Tables 14-15) and GraphSAGE [1] (Tables 16-17), in Appendix F and shown the effectiveness of WGTL on Cora-ML and Polblogs under mettack and nettack. We observe that incorporating WGTL into all of the backbones improves their corresponding performances under a range of perturbation rates. The following table summarizes the %improvement in mean accuracy with respect to the corresponding backbone models:
>
> $$
> \\begin{array}{|c|c|c|c|}
> \\hline
>  & & \\% \text{improvement on Cora-ML} & \\% \text{improvement on Polblogs}  \\\\
> \hline
> GAT+WGTL & Mettack & 0.1-13.8 & 0.6-50.2 \\\\
> GraphSAGE+WGTL & Mettack &       3.2-42.6   &   1.5-40.8 \\\\
> GAT+WGTL    	& Nettack &  0.7-1.1     &   0.14-1.5 \\\\
> GraphSAGE+WGTL & Nettack &    3-4   &    0.4-1.7 \\\\
> \\hline
> \\end{array}
> $$
> [(URL to Tables 14-15)](https://drive.google.com/file/d/1315msJ6yJGCoKncR2Fd6sWimv3tSVj5n/view?usp=sharing)
>
> [(URL to Tables 16-17)](https://drive.google.com/file/d/138WvQLB6ieS-fojA9D1brAKc3NnbSm1H/view?usp=sharing)
>
> > (3) Experiments with WGTL on heterophilic graphs
>
> As a response to this question, we refer the reviewer to point **(c) Comparison with  H${_2}$GCN [3]** of the answer to question (1), and Appendix F.3 in the revised draft.
>
> > (4) Positioning “Related Works” after “Introduction”
>
> Thank you for this suggestion. We have moved the "Related Works" after the "Introduction" section in the revised manuscript.

---

> ### Author Response · Authors · 2023-11-20
> **Response to Technical Questions (2/2)**
>
> References:
>
> [1]  Will Hamilton, Zhitao Ying, and Jure Leskovec. Inductive representation learning on large graphs.
> Advances in neural information processing systems, 30, 2017.
>
> [2] Xiang Zhang and Marinka Zitnik. Gnnguard: Defending graph neural networks against adversarial
> attacks. Advances in neural information processing systems, 33:9263–9275, 2020
>
> [3] Jiong Zhu, Yujun Yan, Lingxiao Zhao, Mark Heimann, Leman Akoglu, and Danai Koutra. Beyond
> homophily in graph neural networks: Current limitations and effective designs. Advances in neural
> information processing systems, 33:7793–7804, 2020
>
> [4] Jiong Zhu, Junchen Jin, Donald Loveland, Michael T Schaub, and Danai Koutra. How does
> heterophily impact the robustness of graph neural networks? theoretical connections and practical
> implications. In Proceedings of the 28th ACM SIGKDD Conference on Knowledge Discovery and
> Data Mining, pp. 2637–2647, 2022
>
> [5] Johannes Gasteiger, Aleksandar Bojchevski, and Stephan Gunnemann. Predict then propagate: Graph
> neural networks meet personalized pagerank. arXiv preprint arXiv:1810.05997, 2018
>
> [6] Petar Velickovic, Guillem Cucurull, Arantxa Casanova, Adriana Romero, Pietro Lio, and Yoshua
> Bengio. Graph attention networks. arXiv preprint arXiv:1710.10903, 2017.

---

> ### Author Response · Authors · 2023-11-22
> **We are eager for your feedbacks**
>
> Dear Reviewer REKj,
>
> Thank you again for the time and effort that you dedicated to providing invaluable feedback on our paper. We would very much appreciate if you could consider updating the scores before the deadline (rapidly approaching). Please let us know If there are any remaining concerns.
>
> Best,
>
> Paper ID 5896 Authors

---

> > ### Comment · Reviewer_REKj · 2023-11-23
> >
> > I appreciate the author for the detailed response. After carefully reading the rebuttal, most of my concerns have been addressed, leading me to update my score. However, some results are still somewhat unconvincing, given their standard deviation and the fact that they were conducted on only two selected datasets. As other reviewers have pointed out, the empirical results the author provided in the rebuttal are necessary and should be thoroughly conducted before submission.

---

### Official Review · Reviewer_UAFj · 2023-11-11

**Soundness:** 2 fair
**Presentation:** 2 fair
**Contribution:** 2 fair
**Rating:** 5
**Confidence:** 4

**Summary:**

Paper proposes to use an approximation to Vietoris Rips complex, the witness complex, with the aim to integrate topological graph features into optimization of GNN, for the purpose of increasing robustness against adversarial attacks. Experiments showing some increase in robustness in several cases are described.

**Strengths:**

* Persistence diagrams are used to propose a topological defense against adversarial attacks in GNN learning.

* The stability of the proposed pipeline is deduced from the known stability properties of persistence diagrams

**Weaknesses:**

* Only one baseline was used for  comparison  in each task. More baseline defense methods involving, in particular, SOTA models, graph attention models and GCN/GNN  with topological regularizers based on standard Vietoris-Rips complexes should be used for comparison.

* Witness complexes although presenting sometimes some advantages in terms of less number of simplexes , are known to suffer from numerous drawbacks:

   * calculation is known to be heavily dependent on the choice of "landmark" points, bringing the instability.

   * sensitivity to parameters, the witness complex setup involves the choice of several hyperparameters, such as the number of landmarks or the epsilon in epsilon-net etc

  * computational complexity, the complex is made smaller but the construction of the complex, i.e. the choice of simplices and their witnesses etc,  becomes more computationally expensive

   * lack of functoriality, the relations between results of calculations in different situations are more difficult to establish.

  The paper mentions some of these concerns, but does not explain really convincingly how to overcome them.

* In particular, it is not explained how to make the crucial choice  concerning  the number of landmarks for the pipeline to work.

* Also,  it is not clear why the standard vietoris-rips complexes, via  GPU acceleration, could not be used instead, to solve the described defense tasks.

* The formulations of the theoretical results are not very clearly stated.

* In the description of the pipelines, in experiment details, in the statements or the proofs of theoretical results, the dimensions of the computed persistence diagrams are not specified.

* The reported computational complexity of the pipeline is not accurate. For example it does not include the complexity of the geodesic distance on the graph.



Below are some specific remarks:

abstract:  "against of" -> against

page 2 "complementary information" - complementary to what? it is not very clear

page 3 "is asymmetric matrix A" -> a symmetric

page 3 "For unweighted graphs we get" -> For unweighted graphs we set

page 3 "increasing $\epsilon$ from 1 to" -> increasing $\epsilon$ from 0 to

page 3 "$\mathcal{G}_{\alpha}$, consisting of only paths with length more than $\alpha$"-> only edges with length more than $\alpha$

page 3  with such definition of $\mathcal{G}_{\alpha}$,
 all the inclusions of alpha-indexed subgraphs or complexes in the paper must be reversed :
 for ${\alpha_1}\leq{\alpha_2}$ the inclusion of the corresponding subgraphs goes in the opposite direction.

page 3 "There are multiple ways to compute simplicial complex"- what is meant by "compute" here? perhaps define or construct ?

page 4 "The weak witness complex ... of the graph... with respect to the landmark set" - a verb is missing here, which makes the definition not very clear

page 5 in Component II, what is  $\Theta^{(0)}$ ?

page 5 in Component II there seems to be a misprint in  $Z^(_{G}0)$

page 6 "the persistence diagram of the auxilary graph reconstructed from transformer output" - what is this auxilary graph? it is not clearly explained

page 6 The reference arXiv:2109.04825 which studied the persistence diagrams of transformer attention graphs is seemingly relevant here

page 6 "is is stable" -> is stable

page 6 what is $A(\mathcal{G})$ in Proposition 3?

page 9 when the standard PH algorithm is mentioned, and in the related works, a reference is missing : Barannikov, S. (1994). The framed Morse complex and its invariants. Advances in Soviet Mathematics, 21, 93-116, where the canonical forms=persistence barcodes were first introduced and the algorithm for their calculation was first described.

page 9 "the homologically persistent graph skeleton" - what is meant by this?

**Questions:**

Why  the standard Vietoris-Rips complexes, with eg subsampling and GPU acceleration, could not be used to solve the described defense tasks?

---

> ### Author Response · Authors · 2023-11-20
> **Response to Technical Questions (1/3)**
>
> We thank the reviewer for his detailed review with minute observations and comments to improve the paper.
>
> > (1) Further comparison with different baselines
>
> In the revised draft, we demonstrate the performance of WGTL with different GNN backbones (for both graphs with homophily and heterophily), and in comparison with an existing defense mechanism. We elaborate the results below.
>
> (a) **Existing Defenses for GNNs:** In Appendix F.2, we have compared WGTL with two SOTA defense methods in the paper: ProGNN (Table 5) and GNNGuard [1] (Tables 18-19). With respect to ProGNN (with GCN backbone) SOTA, our method ProGNN+WGTL gains 0.68% - 4.96% of relative improvements on Cora-ML and Citeseer under mettack. On Cora-ML under mettack, our method GCN+GNNGuard+WGTL yields more than 2.25%, 1.12%, and 1.85% relative improvements with respect to GCN, GCN+GNNGuard (SOTA defense), and GCN+WGTL, respectively.
>
> [(URL to Table 5)](https://drive.google.com/file/d/13AJoUfgaXf_XHYylovvJVWG97YsoV4u2/view?usp=sharing)
>
> [(URL to Tables 18-19)](https://drive.google.com/file/d/13942jxN8ZntEusr3SgL93LScd2G_PRJ7/view?usp=sharing)
>
> (b) **Different GNN Backbone Architectures:** In addition to GCN, We have added two more backbones, GAT [6] (Tables 14-15) and GraphSAGE [2] (Tables 16-17) in Appendix F, and shown the effectiveness of WGTL on Cora-ML and Polblogs under mettack and nettack. We observe that incorporating WGTL into all of the backbones improves their corresponding performances under a range of perturbation rates. The following table summarizes the %improvement in mean accuracy with respect to the corresponding backbone models:
> $$
> \\begin{array}{|c|c|c|c|}
> \\hline
>  & & \\% \text{improvement on Cora-ML} & \\% \text{improvement on Polblogs}  \\\\
> \hline
> GAT+WGTL & Mettack & 0.1-13.8 & 0.6-50.2 \\\\
> GraphSAGE+WGTL & Mettack &       3.2-42.6   &   1.5-40.8 \\\\
> GAT+WGTL    	& Nettack &  0.7-1.1     &   0.14-1.5 \\\\
> GraphSAGE+WGTL & Nettack &    3-4   &    0.4-1.7 \\\\
> \\hline
> \\end{array}
> $$
>
> [(URL to Tables 14-15)](https://drive.google.com/file/d/1315msJ6yJGCoKncR2Fd6sWimv3tSVj5n/view?usp=sharing)
>
> [(URL to Tables 16-17)](https://drive.google.com/file/d/138WvQLB6ieS-fojA9D1brAKc3NnbSm1H/view?usp=sharing)
>
> (c) **Performance for Heterophlic Graphs:** Furthermore, following [4], we incorporated the proposed global topology encoding into H${_2}$GCN [3], a popular method proposed to handle heterophilic graphs. We experiment with snap-patents, a strongly heterophilic graph under mettack. In Table 20 of Appendix F3, we have shown that H${_2}$GCN+WGTL improves the robustness of H2GCN by up to 4%.
>
> Note that the most adversarially robust method on this dataset, APPNP by [5], has been shown to have an accuracy of $27.76\%$ under 20\% perturbation (c.f. Table 3, [4]). Improving on that, we observe that H${_2}$GCN+WGTL achieves $28.21\%$ average accuracy under $20\%$ perturbation.
>
> $$
> \\begin{array}{|c|c|c|c|c|c|c|}
> \\hline
>  \text{Perturbation rates=>} & 0\\% & 5\\% & 10\\%  & 15\\% & 20\\% & 25\\% \\\\
> \hline
> H_2GCN & 27.71\pm 0.86  	& 27.55 \pm 0.19  & 28.62 \pm 0.38 & 28.40 \pm 1.38 & 27.77 \pm 0.30 & 27.45 \pm0.89  \\\\
> H_2GCN+WGTL & 27.72 \pm 0.85 & 28.66 \pm 0.1.68  & 28.79 \pm 1.0 & 28.45 \pm 0.61 & 28.71 \pm 0.66 & 27.90 \pm 0.84 \\\\
> \\hline
> \\end{array}
> $$
>
> This table is included in the paper as **Table 20**.
>
> (d) **Comparison with Vietoris-Rips based encodings:** In Appendix E.2, we have implemented GCN + VRGTL, where instead of encoding witness topological features, we have encoded the Vietoris-Rips features.  Table 12 compares them on Cora-ML and Citeseer under mettack. We observe that both models have a comparable accuracy. However, Table 13 shows that Vietoris-Rips topological features take significantly more time than Witness feature computation. Hence, GCN+VRGTL does not bring much of a benefit at the expense of significantly higher computational resources.
>
> [(URL to Tables 12-13)](https://drive.google.com/file/d/13IFvmsAeIA7J6X3ShUIT9oWxa-7jU0R0/view?usp=sharing)

---

> ### Author Response · Authors · 2023-11-20
> **Response to Technical Questions (2/3)**
>
> >  (2) Discussions on different aspects of Witness complex construction
>
> Here, we elaborate on four points regarding different aspects of Witness complex construction and how we address in this work the four drawbacks mentioned by the reviewer.
>
> a.    The landmark selection heuristic employed is deterministic because we select the top 5% highest-degree nodes as landmarks. Since there is no randomness in such deterministic selection, there is no instability in the final performance.
>
> b.    The accuracy is indeed dependent on the number of landmarks. Figure 3 (Appendix E) shows that with an increased number of landmarks, the accuracy increases.
>
> c.    Indeed, with an increased number of landmarks, the computational cost increases, as we have shown in Figure 3. However, Table 13 indicates that even after such an increase, the computation time is still substantially less than that of Vietoris-Rips. Furthermore, Figure 3 (Appendix E) shows that the accuracy increases, becoming closer to what is obtained by GCN+VRGTL.
>
> d.    It is not clear what the reviewer meant by “different situations”; hence, we consider the following two scenarios:
>
>  1)    The relation between persistence diagrams induced by different sets of landmarks of the same cardinality: [8] show that irrespective of the choice of landmarks, if the landmarks are $\epsilon$-net of the graph $G = (V, E)$, it produces a $(3log3 + 2\epsilon)$-approximations to Vietoris-Rips complex of $G$. Let us assume two sets of landmarks  $L$, $L’$ such that $|L| = |L’|$ and both are $\epsilon$-net of the graph $G = (V, E)$. Then, both choices give us the same approximation error w.r.t Vietoris-Rips.
>
>  2)    The relations between encoding induced by different sets of #landmarks of the same cardinality:  Proposition 2 shows that the encoding $Z_{WGTL}$ is stable under adversarial perturbation. The guarantee is dependent on $\epsilon$ but does not depend on the particular choices of landmarks. Let us assume two sets of landmarks  $L$, $ L’$ such that $|L| = |L’|$ and both are $\epsilon$-net of the graph $G = (V, E)$. Then, both choices of the landmark set give us the same stability guarantee in terms of encoding $Z_{WGTL}$.
>
> >  (3) Explaining the choice concerning the number of the landmarks and its impact
>
> The choice of the number of landmarks is indeed an important hyperparameter for the pipeline to work. We varied the number of landmarks and made an assessment of their impact on accuracy as well as the increased computation time. Figure 3 (Appendix E) shows that increasing the number of landmarks indeed slightly increases the accuracy, albeit with the expense of increased computation time. Due to this trade-off, the selection of an optimal number of landmarks is dependent on how much robustness is desired by a user with a given computation time budget. Appendix E.1  includes a more elaborate discussion on this point.
>
> [(URL to Figure 3)](https://drive.google.com/file/d/1307sEUDuvneaBrEOgzaj9dny_7gr83X1/view?usp=sharing)
>
> > (4) Using the standard Vietoris-Rips complexes in the pipeline
>
> Thank you for bringing this interesting point to our attention. The Vietoris-Rips complexes can be used with the proposed pipeline, which is indicative of the flexibility offered by the proposed method.
>
> In Appendix E.2, we have implemented GCN + VRGTL, where instead of encoding witness topological features, we have encoded the Vietoris-Rips features. Table 12 shows that the accuracy is comparable to that of GCN+WGTL while at the cost of a significant increase in computation time (Table 13). In our implementation, we adopted Ripser (SOTA for CPU-based computation) for computing both Vietoris-Rips and Witness topological features. Hence, if a GPU-accelerated implementation,e.g., Ripser++, were to be used, it would have accelerated Vietoris-Rips as well as Witness feature computations.
>
> [(URL to Tables 12-13)](https://drive.google.com/file/d/13IFvmsAeIA7J6X3ShUIT9oWxa-7jU0R0/view?usp=sharing)
>
> > (5) Clarification of theoretical results
>
> We have clarified the theoretical results in the main paper. In addition, we have included in Appendix D a nomenclature of notations (Appendix D.1) and detailed derivations of the proofs to enhance clarity. Please consider the revised draft for the changes.
>
> > (6) Specifying the dimensions used in the persistence diagrams
>
> Thank you for your detailed observation. Our theoretical results hold true once we fix the dimension of the persistence $d \in Z_{\geq 0}$ to be computed and use it throughout the pipeline. In our experiments, we used 0-dimensional persistence. We have added it in Section 5 (paragraph 1) of our updated manuscript.

---

> ### Author Response · Authors · 2023-11-20
> **Response to Technical Questions (3/3)**
>
> > (7) Computational complexity of the Witness complex-based topological features
>
> Thank you for pointing out this issue.
> We have corrected the computational complexity segment in Section 5 as per the following:
>
> Landmark selection (top-$|\mathfrak{L}|$ degree nodes) has complexity $\mathcal{O}(N\log(N))$. To compute witness topological features, one needs to compute (1) landmarks-to-witness distances costing $\mathcal{O}(|\mathfrak{L}|(N+|\mathcal{E}|))$ due to BFS-traversal from landmarks, (2) landmark-to-landmark distances costing $\mathcal{O}(|\mathfrak{L}|^2)$, and finally (3) persistent homology via boundary matrix construction and reduction [7]. Matrix reduction algorithm costs $\mathcal{O}(\zeta^3)$, where $\zeta$ is the number of simplices in a filtration. Overall, the computational complexity of computing witness topological feature on the graph is $\mathcal{O}(|\mathfrak{L}|(N+|\mathcal{E}|)+|\mathfrak{L}|^2+\zeta^3)$.
>
> > (8) Specific remarks
>
> Thank you for pointing out these mistakes/typos. We have addressed them in our updated manuscript.
> Here, we address two specific questions mentioned in this list.
>
> > page 2 "complementary information" - complementary to what? it is not very clear
>
> By complementary information here, we understand more traditional graph summaries that are not extracted using any tools of persistent homology.
>
> > page 9 "the homologically persistent graph skeleton" - what is meant by this?
>
> We have reformulated this sentence as follows: “Another interesting research direction is to investigate the linkage between the attacker's budget, number of landmarks, and topological attacks targeting the skeleton shape, that is, topological properties of the graph induced by the most important nodes (landmarks).”
>
> References:
>
> [1]  Xiang Zhang and Marinka Zitnik. Gnnguard: Defending graph neural networks against adversarial
> attacks. Advances in neural information processing systems, 33:9263–9275, 2020
>
> [2] Will Hamilton, Zhitao Ying, and Jure Leskovec. Inductive representation learning on large graphs.
> Advances in neural information processing systems, 30, 2017.
>
> [3] Jiong Zhu, Yujun Yan, Lingxiao Zhao, Mark Heimann, Leman Akoglu, and Danai Koutra. Beyond
> homophily in graph neural networks: Current limitations and effective designs. Advances in neural
> information processing systems, 33:7793–7804, 2020
>
> [4] Jiong Zhu, Junchen Jin, Donald Loveland, Michael T Schaub, and Danai Koutra. How does
> heterophily impact the robustness of graph neural networks? theoretical connections and practical
> implications. In Proceedings of the 28th ACM SIGKDD Conference on Knowledge Discovery and
> Data Mining, pp. 2637–2647, 2022
>
> [5] Johannes Gasteiger, Aleksandar Bojchevski, and Stephan Gunnemann. Predict then propagate: Graph
> neural networks meet personalized pagerank. arXiv preprint arXiv:1810.05997, 2018
>
> [6] Petar Velickovic, Guillem Cucurull, Arantxa Casanova, Adriana Romero, Pietro Lio, and Yoshua
> Bengio. Graph attention networks. arXiv preprint arXiv:1710.10903, 2017.
>
> [7] Edelsbrunner, Letscher, and Zomorodian. Topological persistence and simplification. Discrete &
> Computational Geometry, 28:511–533, 2002
>
> [8] Naheed Anjum Arafat, Debabrota Basu, and Stephane Bressan. ε-net induced lazy witness complexes
> on graphs. arXiv preprint arXiv:2009.13071, 2020

---

> ### Author Response · Authors · 2023-11-22
> **We are eager for your feedbacks**
>
> Dear Reviewer UAFj,
>
> Thank you again for the time and effort that you dedicated to providing invaluable feedback on our paper. We would very much appreciate if you could consider updating the scores before the deadline (rapidly approaching). Please let us know If there are any remaining concerns.
>
> Best,
>
> Paper ID 5896 Authors

---

> > ### Comment · Reviewer_UAFj · 2023-11-23
> >
> > I acknowledge reading the authors response. Some improvements are incorporated in the text.
> > One of remarks I'm not convinced by is that GPU accelerated implementation will benefit equally the standard Vietoris-Rips complex and its approximation given by the witness complex.
> > Some other issues need further attention, in particular a better explanation of the need for the complicated pipeline involving the topological features of attention graphs constructed over aggregated local and global topological encodings.

---

### Author Response · Authors · 2023-11-20
**General comment and summary of changes**

We thank the reviewers for their valuable feedback to improve our work. We have uploaded a revised draft in which we make some changes to address the concerns and misunderstandings that have been raised.

As a summary, we have:

- tested performance improvement due to WGTL across a range of perturbations (0% to 25%) for two GNN backbones: GAT and GraphSAGE, along with the results for GCN and ProGNN that was included in the previous version. We added the corresponding results and discussions in Appendix F.1. The results show that WGTL improves the performance across perturbations irrespective of the GNN backbone used with it.

- compared performance of WGTL with an existing adversarial defense mechanism GNNGuard, in addition to ProGNN, and reported the results in Appendix F.2. The results show that GCN+WGTL is more robust than GCN+GNNGuard. Additionally, we are able to use WGTL with GCN+GNNGuard, which leads to further robustness and improvement in performance.

- tested the performance of WGTL on heterophilic graph datasets with a compatible GNN backbone, namely H$_2$GCN, and elaborated the results in Appendix F.3. We show that deploying H$_2$GCN+WGTL leads to consistent improvement in performance also for heterophilic graphs and across different perturbation rates.

- compared the performance of WGTL with that of Vietoris-Rips based Graph Topology Encoding (VRGTL) for different graphs and across different perturbation rates. We report the results in Appendix E.2. Our results indicate that using WGTL leads to similar robustness as using VRGTL, while leading to 100x-1000x improvement in computation time.

- studied the impact of the number of landmarks on the performance of WGTL. We demonstrate the run-time vs robustness trade-off exhibited by WGTL due to the different number of landmarks in Appendix E.1.

- revised all the definitions and typos in the main paper, and the theoretical analysis in Appendix D. We have rectified the errors, added a table of notations (Appendix D.1), and explained all the steps of the proofs for clearer exposition.

Below, we address the questions specific to each reviewer in further details.

---

### Meta-Review · Area_Chair_avXp · 2023-12-15

**Metareview:**

This paper introduces the concept of witness complex to integrate topological graph features into optimization of GNN. The objective is to improve robustness against adversarial attacks. Three experts have evaluated the work and raised many issues. Some of these concerns were related to the presentation of the paper, which have been responded by the authors. However, the other concerns are fundamental and affect the significance of the contribution. For example, the concept of witness complex is known to suffer from various drawbacks. Also, the empirical results should be strengthened, and working on only two selected datasets is not sufficient.

**Justification For Why Not Higher Score:**

I reached this decision by evaluating the contributions and novelty of the work, taking into consideration both the reviews and the responses from the authors.

**Justification For Why Not Lower Score:**

N/A

---

### Decision · Program_Chairs · 2024-01-16

Reject